# DC-GEN: POST-TRAINING DIFFUSION ACCELERATION WITH DEEPLY COMPRESSED LATENT SPACE

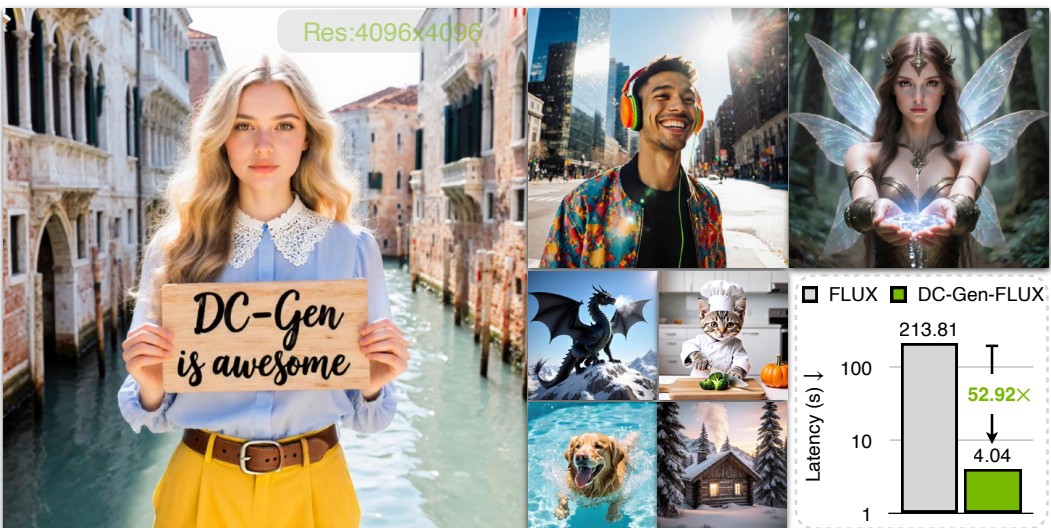

Figure 1: **High-Resolution Image Samples Generated by DC-Gen-FLUX.** DC-Gen-FLUX enables native 4K image generation with about 53× latency reduction over the base model FLUX.1-Krea, measured on an NVIDIA H100 GPU.

## ABSTRACT

Existing text-to-image diffusion models excel at generating high-quality images, but face significant efficiency challenges when scaled to high resolutions, like 4K image generation. While previous research accelerates diffusion models in various aspects, it seldom handles the inherent redundancy within the latent space. To bridge this gap, this paper introduces DC-Gen, a general framework that accelerates text-to-image diffusion models by leveraging a deeply compressed latent space. Rather than a costly training-from-scratch approach, DC-Gen uses an efficient post-training pipeline to preserve the quality of the base model. A key challenge in this paradigm is the representation gap between the base model's latent space and a deeply compressed latent space, which can lead to instability during direct fine-tuning. To overcome this, DC-Gen first bridges the representation gap with a lightweight embedding alignment training. Once the latent embeddings are aligned, only a small amount of LoRA fine-tuning is needed to unlock the base model's inherent generation quality. We verify DC-Gen's effectiveness on SANA and FLUX.1-Krea. The resulting DC-Gen-SANA and DC-Gen-FLUX models achieve quality comparable to their base models but with a significant speedup. Specifically, DC-Gen-FLUX reduces the latency of 4K image generation by 53× on the NVIDIA H100 GPU. When combined with NVFP4 SVDQuant, DC-Gen-FLUX generates a 4K image in just 3.5 seconds on a single NVIDIA 5090 GPU, achieving a total latency reduction of 138× compared to the base FLUX.1-Krea model. ***Code and models will be released.***

# 1 INTRODUCTION

Recent text-to-image diffusion models (Esser et al., 2024; Li et al., 2024e; Chen et al., 2024b; Podell et al., 2023) have achieved remarkable quality, but their slow inference speed remains a major bottleneck, especially for generating high-resolution images like 4K. To address this problem, various acceleration techniques have been explored, such as model quantization (Li et al., 2024c), few-step distillation (Luo et al., 2023), and feature cache (Ma et al., 2024b). However, a critical inefficiency persists: ***the inherent redundancy within the latent space***. Most state-of-the-art text-to-image diffusion models (Lee et al., 2025; Labs, 2024) use a moderately compressed autoencoder to balance reconstruction quality and computational efficiency. For example, FLUX.1 (Labs, 2024) uses a typical $8\times$ compression ratio in its latent autoencoder, representing a 4K image with approximately 65,536 visual tokens. This large number of tokens makes the inference process computationally intensive, requiring over three minutes to generate a single 4K image on an NVIDIA H100 GPU, as shown in Fig. 1.

To address the inherent redundancy in the visual latent space, the DC-AE series (Chen et al., 2024c; 2025b) has demonstrated the potential of deep compression. DC-AE, with $32\times$ and $64\times$ spatial compression ratios, achieves reconstruction quality comparable to less-compressed latent autoencoders (*e.g.*, $8\times$). Integrating such deeply compressed autoencoders into existing diffusion models could lead to substantial efficiency gains. However, building a high-quality text-to-image diffusion model from scratch using a deeply compressed autoencoder is not currently practical. The full training process requires prohibitively computational costs and extensive data efforts. Consequently, the potential benefits of using a deeply compressed latent space to accelerate text-to-image diffusion models remain largely underexplored.

To bridge this gap, we introduce DC-Gen, an effective paradigm that integrates a pretrained text-to-image diffusion model with a deeply compressed autoencoder using cost-efficient post-training. Simply replacing the original autoencoder with a deeply compressed one in a pretrained diffusion model leads to a representation gap between the two latent spaces, causing instability in subsequent fine-tuning (Fig. 3). To solve this problem, we introduce a lightweight ***embedding alignment training*** stage (Sec. 3.3) in DC-Gen, which provides a good starting point for further training. Building on this aligned embedding space, we only need lightweight LoRA fine-tuning to recover the base model's inherent knowledge. With DC-Gen, we can accelerate existing diffusion models using deeply compressed autoencoders while largely preserving their original generation quality.

To verify the effectiveness and broad applicability of DC-Gen, we apply it to two distinct text-to-image diffusion models: SANA (Xie et al., 2024; 2025) and FLUX.1-Krea (Lee et al., 2025). The derived DC-Gen-SANA and DC-Gen-FLUX demonstrate remarkable efficiency improvements while maintaining quality comparable to their respective base models (Table. 2). For instance, DC-Gen-FLUX achieves an approximate $53\times$ latency reduction for 4K image generation compared to the base model FLUX.1-Krea, as demonstrated in Fig. 1. This is all accomplished with a post-training cost of only 40 H100 GPU days.

The contribution of DC-Gen are summarized as follows:

- We introduce DC-Gen, an effective diffusion acceleration paradigm. It connects pre-trained diffusion models with a deeply-compressed latent space through cost-efficient post-training.
- We introduce lightweight embedding alignment training to bridge the latent representation gap when changing the latent space. This stabilizes subsequent fine-tuning and preserves the base model's knowledge.
- We validate DC-Gen's effectiveness on two distinct text-to-image diffusion architectures. With DC-Gen, we can export a series of models for the community that have similar quality to their base models but are significantly more efficient for high-resolution image generation.

# 2 RELATED WORK

## 2.1 DIFFUSION MODEL ACCELERATION

A well-known drawback of diffusion models is their inefficient sampling process, which has motivated numerous works to accelerate it. One series of works focuses on training-free acceleration,

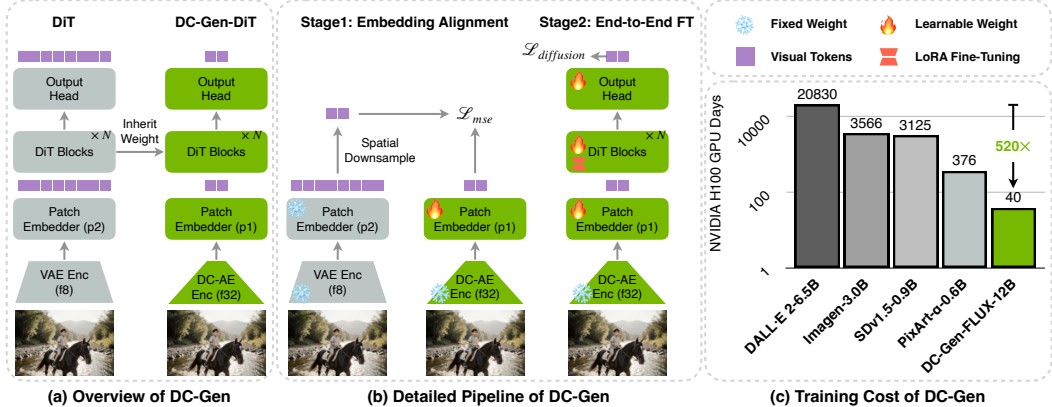

Figure 2: **Illustration of DC-Gen.** (a) DC-Gen accelerates diffusion models by integrating pre-trained diffusion models with deeply compressed autoencoders (*e.g.*, DC-AE (Chen et al., 2024c)). (b) DC-Gen introduces an embedding alignment stage to mitigate discrepancies between embedding spaces, thereby preventing training instability and preserve base model's knowledge. Full details are provided in Table. 7. (c) DC-Gen achieves drastic reductions in training cost relative to training from scratch. DC-Gen-FLUX-12B requires only 40 H100 GPU days, representing a 520× reduction compared to DALL·E 2-6.5B.

including the design of improved ODE solvers (Song et al., 2021; Lu et al., 2022a;b; Zheng et al., 2023; Zhang & Chen, 2023; Zhang et al., 2023; Zhao et al., 2024b), model quantization (Li et al., 2023; 2024c; He et al., 2024; Fang et al., 2024; Zhao et al., 2024a), and efficient computational process (Ma et al., 2024b; Shih et al., 2024; Tang et al., 2024). Another series of works requires post-training for acceleration, like few-step distillation (Luo et al., 2023; Yin et al., 2024a;b) and guidance distillation (Meng et al., 2023; Salimans & Ho, 2022). In addition to accelerating pre-trained models, some works explore efficient architectures (Li et al., 2024d; Liu et al., 2024b; Cai et al., 2024; Ma et al., 2024a; Xie et al., 2024; 2025; Cai et al., 2023) or better inference systems (Frans et al., 2024; Li et al., 2024b; Wang et al., 2024; Xie et al., 2024; Zhou et al., 2025; Geng et al., 2025). DC-Gen falls into the category of post-training acceleration (Gu et al., 2025b), but we investigate a new perspective by accelerating a pretrained diffusion model with a deeply compressed latent space.

## 2.2 AUTOENCODER FOR LATENT DIFFUSION MODELS

Training high-resolution diffusion models directly on raw pixel space (Saharia et al., 2022) is computationally prohibitive. To address this, recent works have explored training latent diffusion models (Rombach et al., 2022; Peebles & Xie, 2023; Bao et al., 2023; Li et al., 2024a; Liu et al., 2024a) on compressed latent spaces learned by autoencoders. State-of-the-art latent diffusion models (Labs, 2024; Lee et al., 2025) typically adopt an 8× compression ratio to balance reconstruction quality and computation efficiency. However, this ratio still results in redundant tokens. To further enhance the latent autoencoder, later works have explored deep compression (Chen et al., 2024a;c; 2025a), improved latent spaces (Yu et al., 2024; Yao et al., 2025; Gu et al., 2024; Chen et al., 2025b; Yao & Wang, 2025; Kouzelis et al., 2025; Skorokhodov et al., 2025; Gu et al., 2025a), or simply increasing latent channel counts (Dai et al., 2023; Labs, 2024; Esser et al., 2024).

While effective deeply compressed autoencoders (Chen et al., 2024c; 2025b) have been developed, training a high-quality text-to-image diffusion model from scratch on these autoencoders remains expensive. Therefore, DC-Gen aims to accelerate pretrained diffusion models with a deeply compressed latent space through a cost-efficient post-training process, instead of costly pretraining.

## 2.3 EFFICIENT AUTOENCODER ADAPTATION

To improve the detail quality of generated images, previous works (Chen et al., 2024b; Rombach et al., 2022) adapt the original VAE to a better version without fully retraining the diffusion model. For example, Chen et al. (2024b) adapts PixArt-$\alpha$-f8c4 to PixArt-$\Sigma$-f8c4, while Rombach et al. (2022) replaces SD-VAE-f8c4 with SD-VAE-v1.5-f8c4. Such adaptations, which typically retain the same VAE architecture and compression ratio, do not involve architectural changes or training

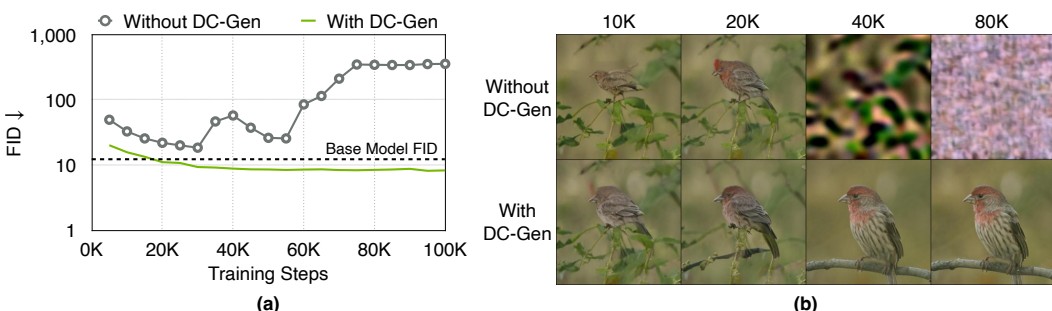

Figure 3: **Challenge in Shifting Latent Space for a Pretrained Model.** When replacing the pretrained DiT's latent space from SD-VAE-f8 to DC-AE-f32, standard fine-tuning fails to reach the base model's performance and is unstable. In contrast, training with DC-Gen is stable and even slightly improves upon the base model's FID. Detailed results are provided in Table. 8 and Fig. 9.

instability when reusing the pretrained model. In contrast, DC-Gen investigates a scenario where the latent autoencoder's structure and compression ratio are different, which often results in training instability due to the representational gap of different latent space.

## 3 METHOD

### 3.1 DC-GEN MOTIVATION AND OVERVIEW

Current state-of-the-art text-to-image diffusion models usually use a moderately compressed latent space, which balances image reconstruction quality with computational efficiency. This approach, however, becomes problematic when generating high-resolution images, such as 2K or 4K. While deeply compressed autoencoders have been developed, training a text-to-image diffusion model on them from scratch is both risky and challenging. This is due to the high cost of pre-training and the limited availability of high-quality image datasets. Consequently, these obstacles hinder the development of text-to-image diffusion models that are both high-quality and efficient.

To address this challenge, this paper introduces DC-Gen, a new paradigm for accelerating a pretrained text-to-image diffusion model by quickly adapting it to a deeply compressed autoencoder via post-training. As the key idea in Fig. 2(a), for a pretrained DiT with an original $8\times$ compressed VAE, we adapt it to a DC-AE with a higher $32\times$ compression ratio. This post-training process inherits the base model's knowledge and prevents costly training from scratch.

The post-trained model, called DC-Gen-DiT, offers several key advantages. (1) DC-Gen-DiT uses fewer tokens to represent an image, which leads to significantly faster inference speeds. (2) DC-Gen-DiT largely inherits the base model's knowledge and semantics without needing access to the original training data or incurring high training costs, as shown in Fig. 2(c). (3) Further fine-tuning on DC-Gen-DiT for applications, such as RL-training or LoRA training, is considerably more efficient than on the base model due to the substantial token reduction.

Despite these notable merits, adapting a diffusion model to a latent space with different structures and compression ratios suffers from instability (Sec. 3.2). To solve this problem, DC-Gen introduces a lightweight embedding alignment training stage before end-to-end fine-tuning (Sec. 3.3). We then demonstrate the effectiveness and broad applicability of DC-Gen's design on two distinct types of diffusion models, SANA and FLUX.1-Krea (Sec. 3.4).

### 3.1.1 PRELIMINARY

In this section, we introduce the key concepts and notations of the latent diffusion model.

- **Compression Ratios.** Latent diffusion models employ two main compression factors to reduce visual tokens: the *latent compression ratio* ($f$) and the *patch size* ($p$). The *latent compression ratio* ($f$) is a hyperparameter of the autoencoder, defined as the ratio of the original image dimensions ($H \times W$) to the latent space dimensions ($H' \times W'$). The *patch size* ($p$), specifically used in models Diffusion Transformer (DiT), is an additional parameter that further reduces the number

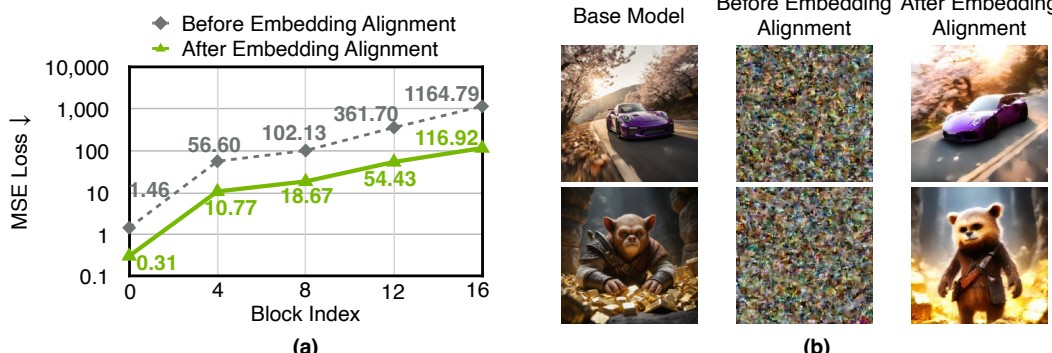

**(a)**                                                                                                         **(b)**

Figure 4: (a) Embedding alignment significantly reduces the per-layer representation gap with the pretrained diffusion model. (b) After embedding alignment, the model can generate images with correct semantics in the new latent space, even without fine-tuning the diffusion model's weights.

of tokens processed by the diffusion model. It achieves this by dividing the latent dimensions into non-overlapping patches of size $p \times p$, with each patch then treated as a single token.

- **Token Counts.** The total number of computational tokens for a diffusion model can be calculated using both compression factors. The formula is as follows:

$$\text{Token Counts} = \left( \frac{H}{f \cdot p} \right) \times \left( \frac{W}{f \cdot p} \right)$$

where $H$ and $W$ represent the original image dimensions. For instance, a latent diffusion model configured as `"f8p2"` uses a latent compression ratio of $f = 8$ and a patch size of $p = 2$. This configuration converts a $1024 \times 1024$ pixel image into 4096 tokens.

- **Components that Binding to the Latent Space.** A diffusion transformer has two core components that bind to different latent spaces. The first is the ***patch embedder***, which transforms the raw latent feature into a patched embedding and map it to the diffusion model's channel. The second is the ***output head***, which maps the latent embedding back to the original latent space's channel and unfolds it. These two components learn a unique mapping and unmapping process between the raw latent feature and the patched embedding. Since they are tied to specific latent autoencoder channels, they can not be reused for different latent spaces in different channels.

## 3.2 CHALLENGE AND ANALYSIS

A key challenge when adapting a pretrained diffusion model to a different autoencoder is the training instability and the catastrophic forgetting of the pretrained model's knowledge.

To examine this effect, we conduct an experiment on class-to-image generation using the ImageNet dataset. We begin with a pretrained DiT model (Peebles & Xie, 2023), originally trained on ImageNet with a stable diffusion VAE (SD-VAE, f8p2) (Rombach et al., 2022). We then replace this VAE with a deeply compressed autoencoder (DC-AE, f32p1) (Chen et al., 2024c), achieving a $4\times$ token compression. Since the two latent autoencoders differ in channel dimensions, we ***randomly initialize*** the patch embedder and the output head that are related to the latent space dimension, as shown in Fig. 2(a). Following this replacement, we fine-tune the new model for 100K steps.

As shown in Fig. 3 (without DC-Gen), the fine-tuning process is unstable, which does not manage to restore the base model's performance. This demonstrates the challenge of getting a pretrained diffusion model to function effectively with a completely new latent space via direct fine-tuning.

## 3.3 DC-GEN PIPELINE

We attribute the training instability primarily to the ***representation gap*** between the two latent spaces. As shown in Fig. 4(a), we examine the per-layer feature distance between the pretrained representations and the newly adapted one. The new representations, encoded with their randomly initialized patch embedder, are biased relative to the pretrained representation, and this error propagates through the layers. To solve this problem, we therefore introduce a lightweight embedding alignment training before performing end-to-end fine-tuning.

| FLUX.1-Krea
(Base Model) | DC-Gen-FLUX
(Full-Tune) | DC-Gen-FLUX
(LoRA-Tune) |
| :---: | :---: | :---: |
| 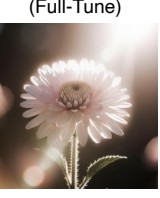 | 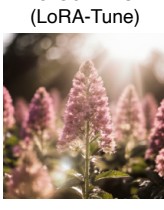 | |

| Method | # Trainable Params↓ | FID↓ | CLIP Score↑ |
| :--- | :---: | :---: | :---: |
| Full-Tune | 11.9B | 49.01 | 26.98 |
| LoRA-Tune | **1.1B** | **48.13** | **27.51** |

(a) Visual comparison of tuning strategies.

(b) Quantitative comparison of tuning strategies.

Figure 5: **Ablation of Tuning Strategies in End-to-End Fine-Tuning.** LoRA tuning better preserves the base model's knowledge during end-to-end fine-tuning and achieves higher quality. Metrics are reported on 1K samples from MJHQ-30K at 512×512 resolution.

**Embedding Alignment.** The core idea of the embedding alignment stage is to pre-align the modules related to the latent space (*i.e.*, the patch embedder and the output head). This provides a good starting point for further fine-tuning.

As illustrated in Fig. 2(b), we denote the un-flattened patch embedding encoded with the pretrained patch embedder as $\mathbf{e} \in \mathbb{R}^{H \times W \times D}$, and the embedding encoded with the randomly initialized patch embedder ($\phi$) as $\mathbf{e}_\phi \in \mathbb{R}^{H' \times W' \times D}$. Here, $D$ is the hidden dimension of the diffusion model. First, we spatially downsample the pretrained patch embedding $\mathbf{e}$ to match $\mathbf{e}_\phi$, which we denote as $\mathbf{e}'$. Then, we update the new patch embedder ($\phi$) to minimize the feature distance between $\mathbf{e}_\phi$ and $\mathbf{e}'$:

$$\mathcal{L}_{mse} = \|\mathbf{e}_\phi - \mathbf{e}'\|_2^2. \tag{1}$$

After training the patch embedder, the output head remains randomly initialized. To further align the output representation, we jointly fine-tune both the patch embedder and the output head for a few steps while keeping the diffusion model's weights fixed.

Following this embedding alignment stage, we visualize the per-layer representation gap in Fig. 4(a) and find that the gap relative to the pre-trained representations is substantially reduced. As shown in Fig. 4(b), after alignment, the model can already generate reasonable images in the new latent space without fine-tuning the diffusion model's weights.

**End-to-End Fine-Tuning.** With embedding alignment, we already have a well-aligned patch embedder and output head. We then jointly finetune them with the diffusion model's weights using the standard flow-matching objective (Lipman et al., 2022; Liu et al., 2022) to further accommodate the new latent space.

Given a image latent sample $\mathbf{x}_1$, we sample a timestep $t \in [0, 1]$ and a noise sample $\mathbf{x}_0$ to construct a training sample $\mathbf{x}_t = (1 - t)\mathbf{x}_0 + t\mathbf{x}_1$. The ground truth velocity is given by $\mathbf{v}_t = \mathbf{x}_1 - \mathbf{x}_0$. Given the input $\mathbf{x}_t$ and text condition $c$, we train our model ($\theta$) to predict the velocity $\mathbf{v}_t$ by minimizing the following objective:

$$\mathcal{L}_{fm} = \mathbb{E}_{t, \mathbf{x}_0, \mathbf{x}_1} \left[ \|\mathbf{v}_\theta(\mathbf{x}_t, c, t) - \mathbf{v}_t\|_2^2 \right]. \tag{2}$$

It is worth noting that after the embedding alignment stage, the model is already capable of generating semantically correct images. Therefore, we only need LoRA-tuning in end-to-end fine-tuning stage to accommodate the new latent space. This strategy largely preserves the original model's knowledge while being more efficient than full-tuning, as demonstrated in Fig. 5.

## 3.4 DC-GEN APPLICATION

### 3.4.1 DC-GEN-SANA

SANA-1.5 (Xie et al., 2025) is a linear diffusion transformer model designed for efficient text-to-image generation. Building on DC-AE-f32 (Chen et al., 2024c), SANA-1.5 has 1.6B and 4.8B variants. We apply DC-Gen to the both variants to adapt them to DC-AE-f64. The resulting DC-Gen-SANA model achieves a token reduction of 4× compared to the base model.

### 3.4.2 DC-GEN-FLUX

FLUX (Labs, 2024) is a state-of-the-art text-to-image diffusion model that has 12B parameters. Among its variants, we choose FLUX.1-Krea (Lee et al., 2025) as our base model because of its

FLUX.1-Krea (Base Model)     DC-Gen-FLUX (with $\mathscr{L}_{fm}$)     DC-Gen-FLUX (with $\mathscr{L}_{fm}^{guide}$)

(a) Visual comparison of training objectives.

| Method | FID↓ | CLIP Score↑ |
|---|---|---|
| Train with $\mathcal{L}_{fm}$ (Eq. 2) | 48.57 | 27.00 |
| Train with $\mathcal{L}_{fm}^{guide}$ (Eq. 7) | **48.13** | **27.51** |

(b) Quantitative comparison of training objectives.

**Figure 6: Ablation of Training Objectives for Guidance-Distilled Models.** $\mathcal{L}_{fm}^{guide}$ (Eq. 7) corrects the velocity estimation and demonstrates a large improvement in visual quality. The metric is reported on 1K samples of MJHQ-30K 512×512.

superior realism and quality. The FLUX.1-Krea is built on a VAE with an $8\times$ compression ratio, which limits its efficiency when generating high-resolution images. We therefore adopt DC-Gen to integrate FLUX.1-Krea with the DC-AE for improve its efficiency.

It's worth noting that FLUX has only made its guidance-distilled model weights publicly available. Adopting the standard flow-matching objective (Eq. 2) for training the guidance-distilled model will result in a biased estimation of the velocity, as exemplified by the results in Fig. 6(with $\mathcal{L}_{fm}$). To solve this problem, we analyze and provide a correct training objective below.

**Revisiting Guidance Distillation.** The core issue stems from how the guidance-distilled model, denoted as $\mathbf{v}_\eta$, is trained. The goal of guidance distillation (Meng et al., 2023) is to train $\mathbf{v}_\eta$ to mimic the output of classifier-free guidance (CFG). The training objective for this distillation is:

$$\mathcal{L}_{distill} = \mathbb{E}_{w,t,\mathbf{x}_0,\mathbf{x}_1} \left[ \|\mathbf{v}_\eta(\mathbf{x}_t, c, t, w) - \mathbf{v}_\theta^w(\mathbf{x}_t, c, t)\|_2^2 \right]. \tag{3}$$

Here, $w \in [w_{min}, w_{max}]$ represents the guidance scale, and $\mathbf{v}_\theta^w(x_t)$ is the CFG objective:

$$\mathbf{v}_\theta^w(\mathbf{x}_t, c, t) = (1+w)\mathbf{v}_\theta(\mathbf{x}_t, c, t) - w\mathbf{v}_\theta(\mathbf{x}_t, \varnothing, t). \tag{4}$$

Since the publicly available guidance-distilled model $\mathbf{v}_\eta$ is trained to produce this CFG output, simply applying the standard flow-matching objective will be incorrect.

**Correcting the Velocity Estimation.** To address this, we must derive the "raw" velocity $\mathbf{v}_\theta$ from the distilled model $\mathbf{v}_\eta$. We can do so by recognizing that the unconditional velocity, $\mathbf{v}_\theta(\mathbf{x}_t, \varnothing, t)$, can be approximated by the distilled model's output when the condition is empty, i.e., $\mathbf{v}_\eta(\mathbf{x}_t, \varnothing, t, w) \approx \mathbf{v}_\theta(\mathbf{x}_t, \varnothing, t)$. By treating the distilled model's output as the CFG output, we have:

$$\mathbf{v}_\eta(\mathbf{x}_t, c, t, w) \approx (1+w)\mathbf{v}_\theta(\mathbf{x}_t, c, t) - w\mathbf{v}_\eta(\mathbf{x}_t, \varnothing, t, w). \tag{5}$$

To solve for the desired "raw" velocity $\mathbf{v}_\theta(\mathbf{x}_t, c, t)$, we can algebraically rearrange this equation:

$$\mathbf{v}_\theta(\mathbf{x}_t, c, t) \approx \frac{1}{1+w}[\mathbf{v}_\eta(\mathbf{x}_t, c, t, w) + w\mathbf{v}_\eta(\mathbf{x}_t, \varnothing, t, w)]. \tag{6}$$

This expression gives us the corrected velocity estimation, which we denote as $\hat{\mathbf{v}}_\eta = \mathbf{v}_\theta(\mathbf{x}_t, c, t)$. The corrected optimization objective thus becomes:

$$\mathcal{L}_{fm}^{guide} = \mathbb{E}_{w,t,\mathbf{x}_0,\mathbf{x}_1} \left[ \|\hat{\mathbf{v}}_\eta(\mathbf{x}_t, c, t, w) - \mathbf{v}_t\|_2^2 \right]. \tag{7}$$

With the $\mathcal{L}_{fm}^{guide}$, we can enable direct training the guidance-distilled model, as shown in Fig. 6.

## 4 EXPERIMENTS

### 4.1 EXPERIMENTAL SETTINGS

**Class-to-Image Generation.** For class-to-image generation, we conduct experiments on the ImageNet dataset (Deng et al., 2009), utilizing both the 256×256 and 512×512 versions. Our base model is pre-trained DiT-XL (Peebles & Xie, 2023), which is built upon a Stable Diffusion VAE (Rombach et al., 2022) with an $8\times$ compression ratio. To accelerate this model, we adapt it to the DC-AE-f32 latent space to get DC-Gen-DiT-XL, achieving a $4\times$ token compression. We provide detailed training setting in Table. 7.

**Class-to-Image Generation Results on ImageNet**

| Methods | Autoencoder | Training Steps | Throughput (images/s) ↑ | gFID ↓ w/o CFG | gFID ↓ w/ CFG | Inception Score ↑ |
|---|---|---|---|---|---|---|
| **Resolution 256×256** | | | | | | |
| DiT-XL | DC-AE-f32c32 [7] | 3000k | 12.00 | 9.96 | 2.94 | 109.45 |
| DiT-XL | SD-VAE-f8c4 [40] | 7000k | 2.44 | 9.62 | 2.27 | **121.50** |
| DC-Gen-DiT-XL | SD-VAE-f8c4 → DC-AE-f32c32 | 500k | 12.00 | **8.01** | **2.25** | 118.42 |
| **Resolution 512×512** | | | | | | |
| DiT-XL | DC-AE-f32c32 [7] | 3000k | 4.03 | 9.56 | 2.84 | 117.48 |
| DiT-XL | SD-VAE-f8c4 [40] | 3000k | 0.85 | 12.03 | 3.04 | 105.25 |
| DC-Gen-DiT-XL | SD-VAE-f8c4 → DC-AE-f32c32 | 100k | 4.03 | **8.21** | **2.22** | **122.51** |

Table 1: **Evaluation on Class-to-Image Generation.** DC-Gen-DiT-XL achieves better performance to the base model DiT-XL (SD-VAE-f8), while reaching about 4× throughput improvement.

**Text-to-Image Generation Results on MJHQ-30K 1024×1024**

| Models | Autoencoder | Throughput (images/min) ↑ | CLIP Score ↑ | GenEval ↑ |
|---|---|---|---|---|
| LUMINA-Next [62] | - | 13.29 | 26.84 | 0.46 |
| SD3-medium [11] | - | 31.00 | 27.83 | 0.62 |
| Hunyuan-DiT [27] | - | 5.54 | 28.19 | 0.63 |
| PixArt-Σ [6] | - | 44.28 | 28.26 | 0.54 |
| SDXL [39] | - | 16.61 | 29.03 | 0.55 |
| PlayGround [22] | - | 23.25 | 29.13 | 0.56 |
| SANA-1.6B [48] | DC-AE-f32c32 [7] | 110.70 | 29.01 | 0.82 |
| DC-Gen-SANA-1.6B | DC-AE-f64c128 [8] | 435.68 | 28.91 | 0.82 |
| SANA-4.8B [49] | DC-AE-f32c32 [7] | 37.68 | 29.23 | 0.81 |
| DC-Gen-SANA-4.8B | DC-AE-f64c128 [8] | 146.23 | 29.03 | 0.84 |
| FLUX.1-Krea-12B | FLUX-VAE-f8c16 | 16.82 | 27.93 | 0.69 |
| DC-Gen-FLUX.1-Krea-12B | DC-AE-f32c32 [7] | 69.37 | 27.94 | 0.72 |

Table 2: **Evaluation on Text-to-Image Generation.** DC-Gen provides consistent efficiency gains on SANA and FLUX.1-Krea, while achieving comparable generation quality to the base model.

**Text-to-Image Generation.** We experiment the DC-Gen on SANA (1.6B and 4.8B versions) and FLUX.1-Krea-12B. For SANA, which is already trained with DC-AE-f32, we replace its latent space with DC-AE-f64, achieving a 4× token reduction. For FLUX.1-Krea, we replace its SD-VAE-f8 with DC-AE-f32 for 512 and 1K resolutions. To further improve efficiency in 2K and 4K image generation, we use DC-Gen to adapt from DC-AE-f32 to DC-AE-f64 for those resolutions.

For both models, we use synthetic dataset generated from the base model to training. We fix the LoRA with a rank and alpha of 256 in end-to-end fine-tuning stage. The remaining detailed training hyper-parameters are provided in Table. 7.

For quantitative evaluation, we use the MJHQ-30K dataset, following common practice (Xie et al., 2024). We report CLIP-Score and GenEval performance for comparison to previous methods.

**Speed Benchmark Setting.** We evaluate the latency and throughput on a single NVIDIA H100 GPU with "`torch.compile`" feature. Latency is measured with batch size of 1 and throughput is measured with maximum achievable batch size within the same memory constraints.

## 4.2 MAIN RESULTS

### 4.2.1 CLASS-TO-IMAGE GENERATION

As shown in Table. 1, DC-Gen-DiT-XL not only outperforms the base DiT-XL model (SD-VAE-f8) but also achieves a throughput improvement of more than 4×. Furthermore, compared to training

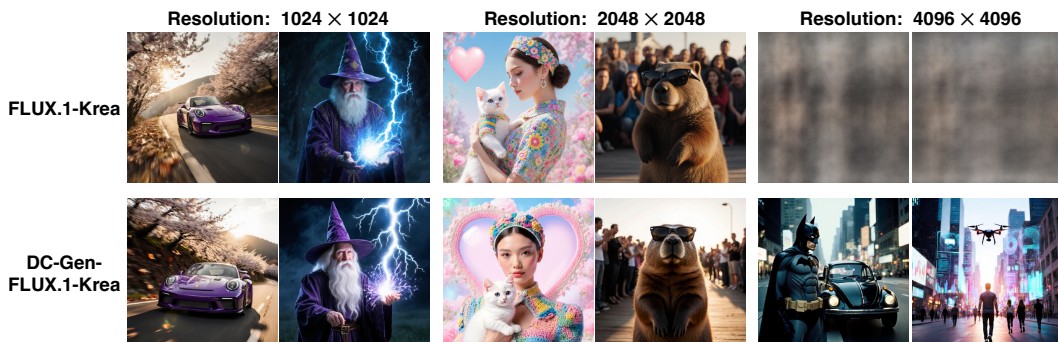

Figure 7: **Qualitative Comparison between FLUX.1-Krea and DC-Gen-FLUX.** DC-Gen-FLUX achieves image generation quality and text alignment on par with the base model, FLUX.1-Krea.

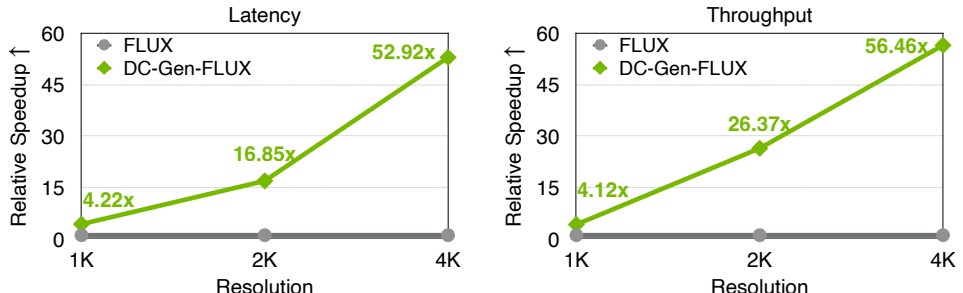

Figure 8: **Speed Comparison Across Different Resolutions.** DC-Gen-FLUX achieves approximately $53\times$ latency reduction and a $56\times$ throughput improvement over the base model at 4K resolution. Detailed quantitative results are provided in Table. 5.

DiT-XL from scratch on the DC-AE-f32, DC-Gen requires significantly less training cost and yields better performance.

### 4.2.2 TEXT-TO-IMAGE GENERATION

**Quantitative Comparison.** As the results summarized in Table. 2, DC-Gen-FLUX and DC-Gen-SANA models achieve similar quantitative results to their respective base models, but with an approximately $4\times$ throughput improvement at 1K resolution.

**Qualitative Comparison.** We demonstrate a qualitative comparison between FLUX.1-Krea and DC-Gen-FLUX across image resolutions from 1K to 4K in Fig. 7 and Fig. 10-Fig. 13. DC-Gen-FLUX shows comparable quality and text alignment to the base model, which demonstrates that we successfully preserve the base model's knowledge when shifting to a more efficient latent space.

Furthermore, the base model FLUX.1-Krea does not natively support 4K image generation, likely due to high training costs that prevent training on 4K resolution. In contrast, DC-Gen-FLUX benefits from its deeply compressed latent space to enable efficient training on high-resolution images, thereby unlocking native 4K generation capability.

**4K-Resolution Image Generation Evaluation.** FLUX.1-Krea does not support native 4K image generation. Besides our DC-Gen, Zhang et al. (2025) provides a state-of-the-art framework for enabling 4K generation. Table. 3 summarizes the comparison on the Diffusion-4K benchmark (Zhang et al., 2025). DC-Gen delivers substantially better efficiency while also achieving higher CLIP scores, higher Aesthetic scores, and noticeably better visual quality (Fig. 16).

**Speed Comparison.** We benchmarked the latency and throughput of FLUX.1-Krea and DC-Gen-FLUX across various resolutions. With higher resolutions, which contain more redundancy, DC-Gen can exploit a deeper autoencoder with a larger compression ratio to further eliminate redundancy (*i.e.*, we use DC-AE-f64 for 2K and 4K resolution). At 4K resolution, DC-Gen achieves about a $53\times$ improvement in latency and a $56\times$ improvement in throughput over the baseline.

| Method | Steps | Latency (s) ↓ | CLIP Score ↑ | Aesthetic Score ↑ |
|---|---|---|---|---|
| FLUX.1-Krea [21] | 20 | 218.81 | - | - |
| Diffusion-4K (FLUX.1-WLF) [55] | 20 | 23.76 | 32.79 | 5.94 |
| | 50 | 59.42 | 33.12 | 6.06 |
| DC-Gen-FLUX.1-Krea | 20 | **4.04** | **35.05** | **6.30** |

Table 3: **4K-Resolution Image Generation Evaluation on Diffusion-4K [55].**

| Domain | Visual Quality | | | Text Alignment | | |
|---|---|---|---|---|---|---|
| | DC-Gen-FLUX.1-Krea is better | FLUX.1-Krea is better | Equal | DC-Gen-FLUX.1-Krea is better | FLUX.1-Krea is better | Equal |
| General | **47.4%** | 43.0% | 9.6% | **46.4%** | 40.6% | 13.0% |
| Person | **46.8%** | 44.6% | 8.6% | **43.4%** | 43.0% | 13.6% |
| Anime | **48.0%** | 44.6% | 7.4% | **47.4%** | 41.8% | 10.8% |
| All | **47.4%** | 44.1% | 8.5% | **45.7%** | 41.8% | 12.5% |

Table 4: **Human Evaluation Results.** We conducted a human evaluation on Amazon Mechanical Turk. DC-Gen-FLUX.1-Krea achieved a slightly higher win rate than FLUX.1-Krea in both visual quality and text alignment.

### 4.3 HUMAN EVALUATION

We conduct a human evaluation to compare DC-Gen-FLUX.1-Krea and FLUX.1-Krea on 1K-resolution image generation. To ensure unbiased assessment, we use Amazon Mechanical Turk, a crowdsourcing platform. Our evaluation covers three common T2I use cases—general, person, and anime. Each domain includes 25 test prompts, resulting in 75 total prompts. For each prompt, we generate an image pair using DC-Gen-FLUX.1-Krea and FLUX.1-Krea and present the pair to 20 users. Sample images are shown in Fig. 15. Users answer two questions: which image has higher visual quality, and which image aligns more closely with the prompt. The interface is illustrated in Fig. 14. Table. 4 summarizes the results. DC-Gen-FLUX.1-Krea achieves comparable visual quality and text alignment to FLUX.1-Krea, with a slightly higher win rate across all domains.

## 5 CONCLUSION

We have presented DC-Gen, a new framework that accelerates pretrained diffusion models through the use of a deeply compressed latent space. Our solution overcomes the challenge of training instability with a simple yet effective embedding alignment step, followed by LoRA finetuning to preserve the model's knowledge when changing latent space. The effectiveness of DC-Gen is demonstrated through our experiments with SANA and FLUX.1-Krea. The resulting DC-Gen-SANA and DC-Gen-FLUX models match their base models' performance while achieving significant efficiency improvements. DC-Gen paves the way for efficiently scaling the training and application of diffusion models to higher resolutions.

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

# A APPENDIX

## A.1 THE USE OF LLM

All key intellectual work on this project, including the research idea, experiment design, execution, data analysis, and the creation of figures and tables, is complete without using any LLMs. We only use an LLM for polishing during the final manuscript writing.

## A.2 ADDITIONAL EXPERIMENTAL RESULTS

### A.2.1 SPEED BENCHMARK

We report speed benchmarks on the H100 GPU in Table. 5. Notably, DC-Gen is complementary to other acceleration techniques such as model quantization. On the NVIDIA 5090 GPU, combining DC-Gen-FLUX with NVFP4 SVDQuant (Li et al., 2024c) yields further gains, as shown in Table. 6. Specifically, this combination reduces end-to-end inference latency by up to $138\times$. The significant improvement results from DC-Gen reducing token redundancy and SVDQuant eliminating CPU offloading, which is otherwise necessary to fit both the DiT and text encoder into GPU memory.

| (a) Latency (second per image) | | | | (b) Throughput (images per minute) | | | |
|---|---|---|---|---|---|---|---|
| **Method** | **Resolution** | | | **Method** | **Resolution** | | |
| | **1K** | **2K** | **4K** | | **1K** | **2K** | **4K** |
| FLUX.1-Krea [21] | 4.65 | 23.76 | 213.81 | FLUX.1-Krea [21] | 16.82 | 2.51 | 0.28 |
| DC-Gen-FLUX | 1.10 | 1.41 | 4.04 | DC-Gen-FLUX | 69.37 | 66.20 | 15.81 |
| Speedup | 4.22× | 16.85× | 52.92× | Speedup | 4.12× | 26.37× | 56.46× |

Table 5: **Speed Comparison Across Different Resolution on a single NVIDIA H100 GPU.** We visualize the corresponding results in Fig. 8.

| (a) DiT Latency (second per step) | | | | (b) End-to-end Latency (second per image) | | | |
|---|---|---|---|---|---|---|---|
| **Method** | **Resolution** | | | **Method** | **Resolution** | | |
| | **1K** | **2K** | **4K** | | **1K** | **2K** | **4K** |
| FLUX.1-Krea [21] | 0.42 | 2.12 | 19.22 | FLUX.1-Krea [21] | 22.33 | 56.91 | 486.96 |
| DC-Gen-FLUX | 0.07 | 0.14 | 0.42 | DC-Gen-FLUX | 15.17 | 16.38 | 22.33 |
| DC-Gen-FLUX + SVDQuant | 0.03 | 0.05 | 0.16 | DC-Gen-FLUX + SVDQaunt | 0.65 | 0.99 | 3.52 |
| Total Speedup | 14.00× | 42.40× | 120.13× | Total Speedup | 34.35× | 57.48× | 138.34× |

Table 6: **Speedup of DC-Gen-FLUX with SVDQuant (Li et al., 2024c) on NVIDIA 5090 GPU.**

### A.2.2 QUALITATIVE RESULTS

**Visual Comparison to Base Model.** We provide a qualitative comparison between DC-Gen and its base models in Fig. 10 and Fig. 11. Our findings show that DC-Gen-SANA retains the generation quality and multilingual capabilities of SANA while increasing throughput by approximately $4\times$. Similarly, DC-Gen-FLUX, DC-Gen-FLUX inherits the superior realism of FLUX.1-Krea and also achieves about a $4\times$ improvement in throughput.

**Visual Comparison to Prior Model.** We compare our DC-Gen-FLUX model to previous state-of-the-art text-to-image diffusion models in Fig. 12. FLUX.1-Krea is noted for its superior realism and text rendering capabilities but is hindered by a lower throughput. Our DC-Gen-FLUX successfully inherits these qualities while providing a significant speedup over FLUX.1-Krea, resulting in the best throughput among the compared models.

**Visualization of 4K Image Generation Results.** We show the 4K image generation results of DC-Gen-FLUX in Fig. 13. While FLUX.1-Krea cannot generate 4K images (as shown in Fig. 7), DC-Gen-FLUX benefits from its efficient training on high-resolution images, which unlocks the native 4K image generation ability.

## A.3 HYPERPARAMETERS

We provide the detailed training hyper-parameters for main experiments in Table. 7.

| Training Stage | Hyper-parameter | DiT [38], Class-to-Image | SANA [49], Text-to-Image | FLUX [21], Text-to-Image |
|---|---|---|---|---|
| Patch Embedder Alignment | learning rate | 1e-4 | 1e-4 | 1e-4 |
| | warmup steps | 0 | 0 | 0 |
| | batch size | 64 | 64 | 64 |
| | training steps | 50k | 20k | 20k |
| | optimizer | AdamW, betas=[0.9, 0.999] | AdamW, betas=[0.9, 0.999] | AdamW, betas=[0.9, 0.999] |
| Output Head Alignment | learning rate | 2e-4 | 2e-4 (1.6B) / 1e-4 (4.8B) | 1e-4 |
| | warmup steps | 0 | 0 | 0 |
| | batch size | 1024 | 1024 (1.6B) / 256 (4.8B) | 256 |
| | training steps | 20k | 5k | 5k |
| | optimizer | AdamW, betas=[0.9, 0.999] | AdamW, betas=[0.9, 0.999] | AdamW, betas=[0.9, 0.999] |
| End-to-End Fine-Tuning | learning rate | 2e-4 | 2e-4 (1.6B) / 1e-4 (4.8B) | 1e-4 |
| | warmup steps | 2k | 2k | 2k |
| | training steps | 100k (512px) / 500k (256px) | 150k (1.6B) / 50k (4.8B) | 10k |
| | batch size | 1024 | 1024 (1.6B) / 256 (4.8B) | 256 |
| | optimizer | AdamW, betas=[0.9, 0.999] | AdamW, betas=[0.9, 0.999] | AdamW, betas=[0.9, 0.999] |
| | weight decay | 1e-3 | 1e-3 | 1e-3 |
| | ema | 0.999 | 0.999 | 0.999 |

Table 7: **Training Hyperparameter of DC-Gen.** Output head alignment uses the same loss as end-to-end fine-tuning, with the only difference being that LoRA is not used during this stage.

### A.3.1 Ablation Study

**Embedding Alignment Training.** The key to the success of DC-Gen is its lightweight embedding training, which aligns the two different latent spaces and stabilizes the training. To ablate this key contribution, we perform ablation studies on DiT-XL, SANA, and FLUX.1-Krea. The results, as summarized in Table. 8, show that without alignment training, DiT and SANA suffer from training instability and fail to converge. While FLUX.1-Krea does not train to failure, it achieves inferior results (image blur, noticable artifacts, etc.) compared to alignment training, as visualized in Fig. 9.

**LoRA Tuning.** LoRA is adopted in end-to-end fine-tuning to preserve the pretrained knowledge of the base model. This effect is visualized in Fig. 5 and Fig. 9. Without LoRA tuning, full tuning will be more biased away from the pretrained model.

**Training Objective $\mathcal{L}_{fm}^{guide}$ for Guidance-Distilled Model.** We also visualize the effect of the revised training objective $\mathcal{L}_{fm}^{guide}$ for the guidance-distilled model in Fig. 6 and Fig. 9. Without $\mathcal{L}_{fm}^{guide}$, the training will be biased away from the CFG distribution and demonstrate degraded quality.

(a) Ablation of Embedding Alignment Training on DiT-XL.

| Model and Autoencoder | Training Steps | With Embedding Alignment | Throughput (images/s) ↑ | gFID ↓ w/o CFG | gFID ↓ w/ CFG | Inception Score ↑ |
|---|---|---|---|---|---|---|
| **Class-to-Image Generation Results on ImageNet** | | | | | | |
| **Resolution 256×256** | | | | | | |
| DiT-XL (SD-VAE-f8c4 → DC-AE-f32c32) | 500k | ✗ | 12.00 | 456.10 | 226.73 | 1.00 |
| | | ✓ | 12.00 | 8.01 | 2.25 | 118.42 |
| **Resolution 512×512** | | | | | | |
| DiT-XL (SD-VAE-f8c4 → DC-AE-f32c32) | 100k | ✗ | 4.03 | 344.07 | 243.26 | 1.27 |
| | | ✓ | 4.03 | 8.21 | 2.22 | 122.51 |

(b) Ablation of Embedding Alignment Training on SANA and FLUX.1-Krea.

| Model and Autoencoder | Training Steps | With Embedding Alignment | Throughput (images/min) ↑ | FID ↓ | CLIP-Score ↑ |
|---|---|---|---|---|---|
| **Text-to-Image Generation Results on MJHQ-30K 512×512** | | | | | |
| SANA-1.6B (DC-AE-f32c32 → DC-AE-f64c128) | 150k | ✗ | 1515.31 | 258.50 | 13.72 |
| | | ✓ | 1515.31 | 5.10 | 28.04 |
| SANA-4.8B (DC-AE-f32c32 → DC-AE-f64c128) | 50k | ✗ | 511.72 | 266.41 | 14.54 |
| | | ✓ | 511.72 | 5.18 | 27.92 |
| FLUX.1-Krea-12B (FLUX-VAE-f8c16 → DC-AE-f32c32) | 10k | ✗ | 118.96 | 15.78 | 26.50 |
| | | ✓ | 118.96 | 13.30 | 27.18 |

Table 8: **Ablation Study of Embedding Alignment Training.** Direct fine-tuning without embedding alignment training, suffers from instability and may lose pretrained knowledge from the base model, which leads to inferior performance.

### A.4 Limitation and Future Work

**Limitation.** DC-Gen is a general post-training framework designed to accelerate diffusion models while preserving the original model's knowledge. One limitation of DC-Gen is that its performance is bound by the quality of the pretrained models it uses. Another limitation is that at smaller resolu-

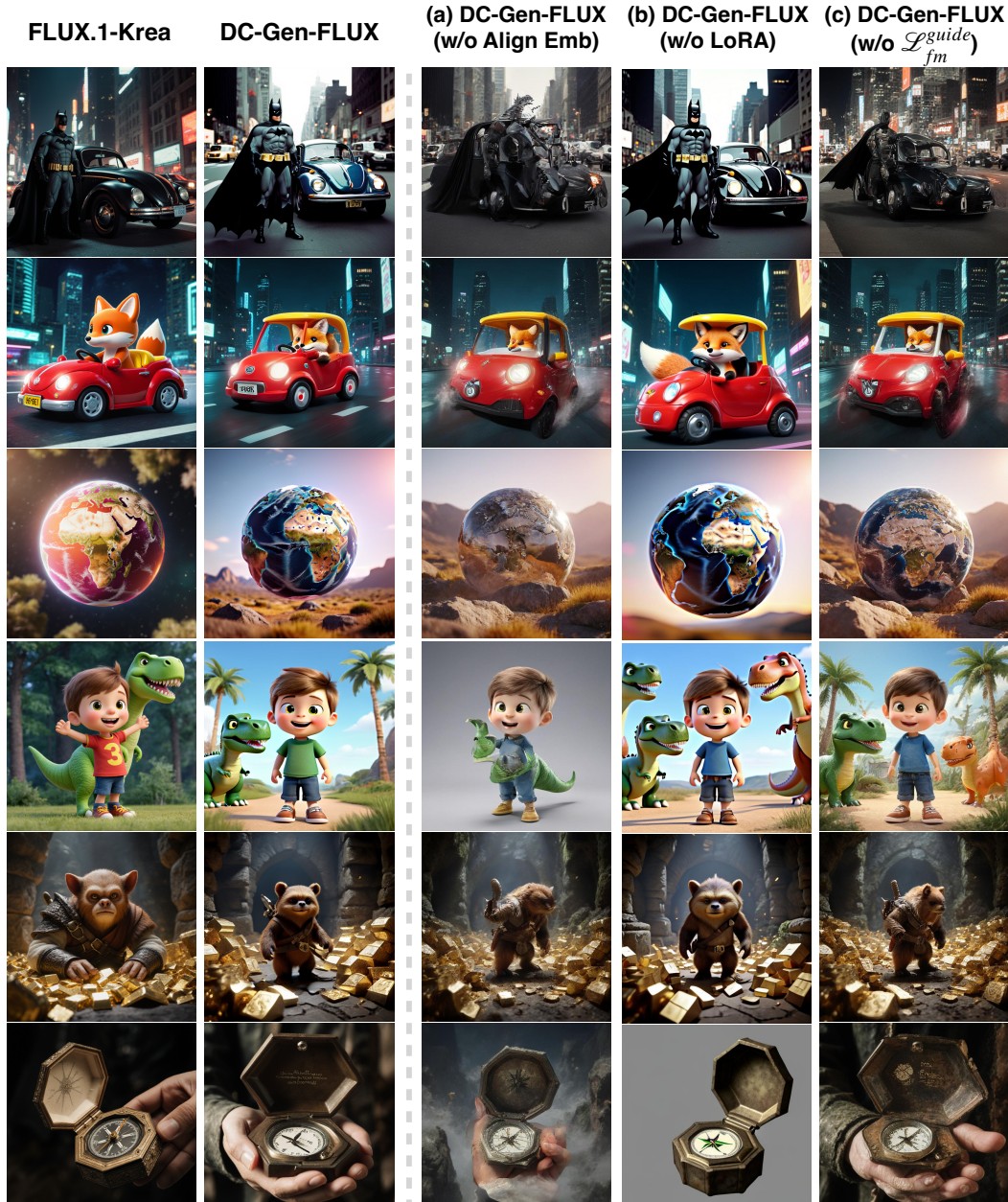

Figure 9: **Visual Ablation of Key Ingredients in DC-Gen**. (a) Without embedding alignment, training is unstable, leading to quality degradation. (b) Without LoRA training, the model loses pretrained knowledge and shows less text alignment. (c) Without the revised training objective for the guidance-distilled model, the model biases away from the CFG distribution and exhibits quality degradation.

tions (*e.g.*, 512), a highly compressed latent space (*e.g.*, f64) can degrade the quality of fine details. However, this issue isn't caused by DC-Gen itself, but rather by the deeply compressed autoencoder. This degradation will be significantly reduced in high-resolution settings.

**Future Work.** An important direction is to apply DC-Gen to video generation, which faces significant efficiency challenges in managing the large token count required for high-resolution video generation. Another interesting direction is to design more effective deeply compressed autoencoders and use DC-Gen as a platform to test and compare them. Since DC-Gen offers a cost-efficient way to integrate a pretrained autoencoder with a pretrained diffusion model, we could easily check the autoencoders' performance to quickly adapt them to the target diffusion model.

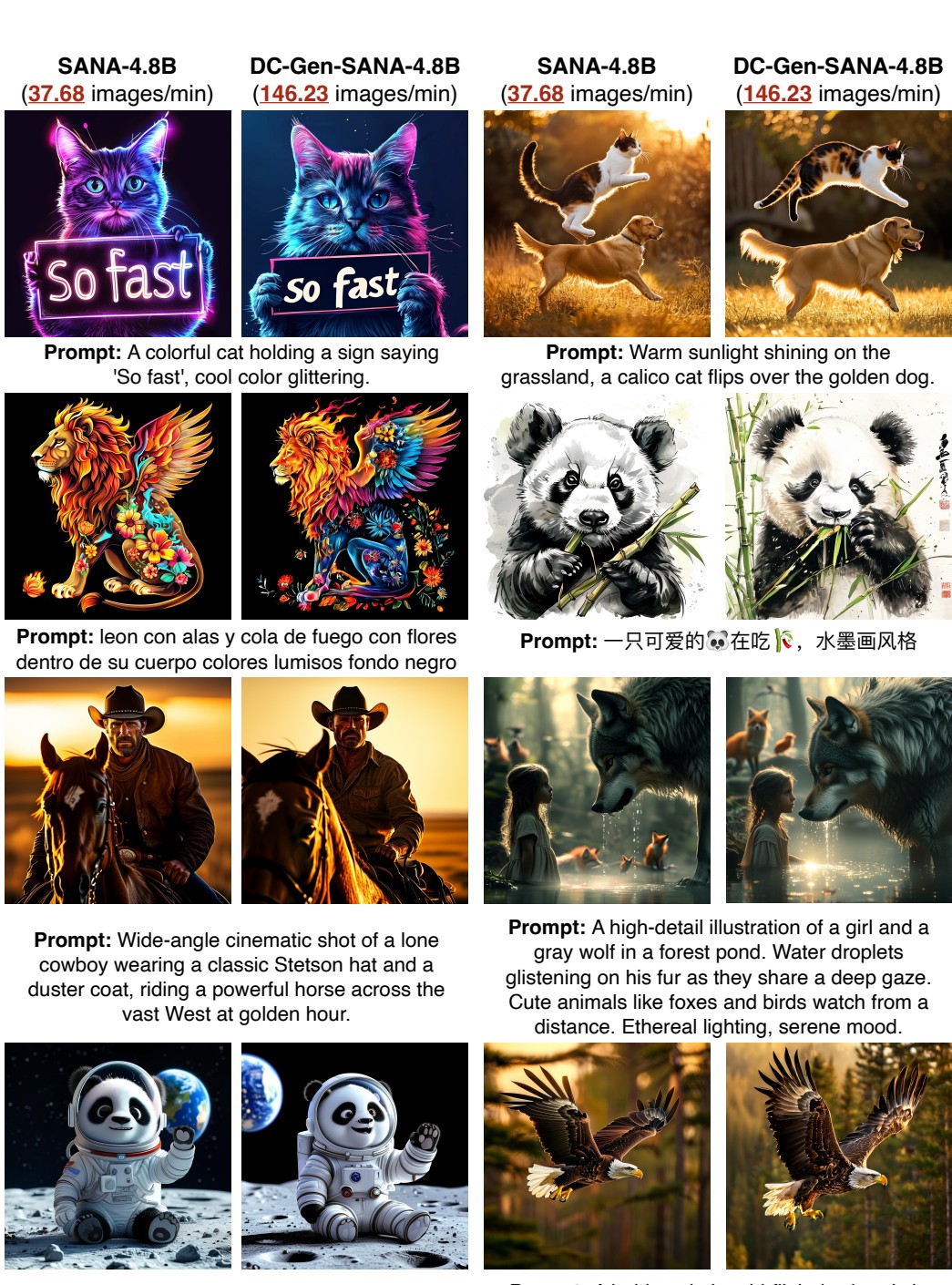

Figure 10: **Samples of SANA-4.8B and DC-Gen-SANA-4.8B on 1024×1024 Resolution.** DC-Gen-SANA preserves the generation quality and multilingual ability of SANA, while achieving an approximately 4× throughput improvement.

| **FLUX.1-Krea** (**17** images/min) | **DC-Gen-FLUX** (**69** images/min) | **FLUX.1-Krea** (**17** images/min) | **DC-Gen-FLUX** (**69** images/min) |
|---|---|---|---|

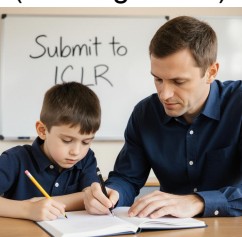 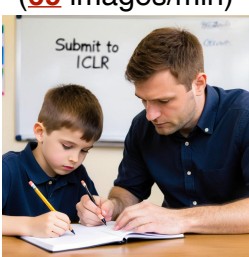 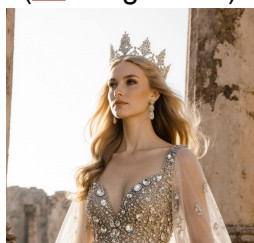 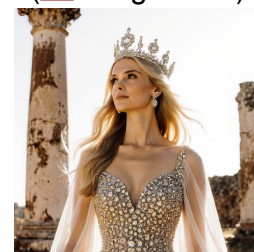

**Prompt:** A man mentors a boy at a table, pointing at a notebook in a classroom with 'Submit to ICLR' on the whiteboard.

**Prompt:** Golden-haired queen in a jeweled crown and a sparkling gown, gazing across sunlit ancient ruins.

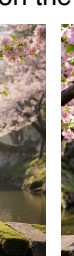 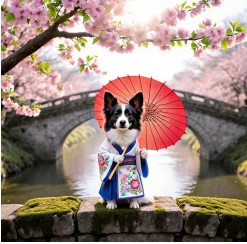 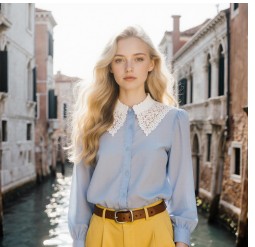 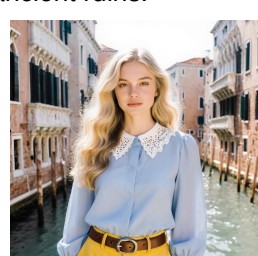

**Prompt:** A mini Border Collie in a traditional Hanfu, holding an oil-paper umbrella on an ancient stone bridge.

**Prompt:** A young woman with wavy blonde hair, wearing a blue blouse and yellow pants, smiling in a sunlit Venice.

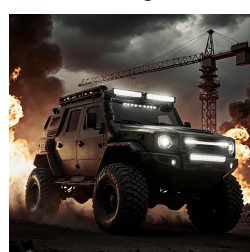 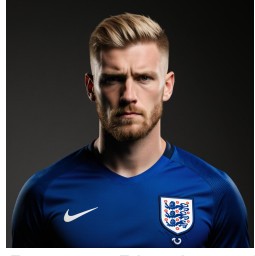 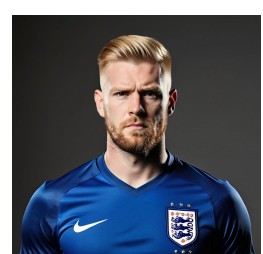

**Prompt:** Futuristic armored off-road vehicle with bright lights driving through chaotic destruction and flames.

**Prompt:** Blond man in a blue England soccer jersey, with an intense gaze, captured in a simple studio portrait.

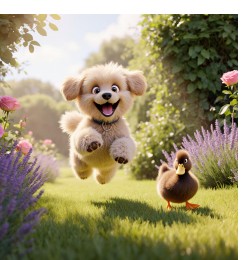 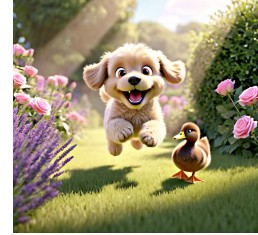 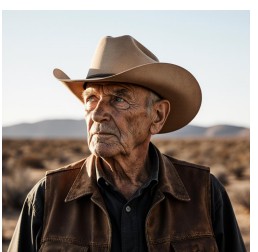 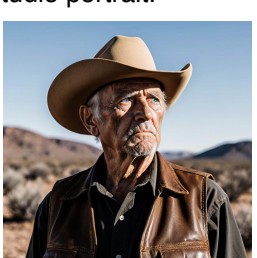

**Prompt:** A whimsical 3D animated Cavapoo playfully chases a duck through a sun-drenched, blooming country garden.

**Prompt:** Rugged old cowboy with a tan hat, looking thoughtfully across a harsh, sunlit desert landscape.

Figure 11: **Samples of FLUX.1-Krea and DC-Gen-FLUX on 1024×1024 Resolution.** DC-Gen-FLUX preserves the generation quality and superior realism of FLUX.1-Krea, while achieving an approximately 4× throughput improvement.

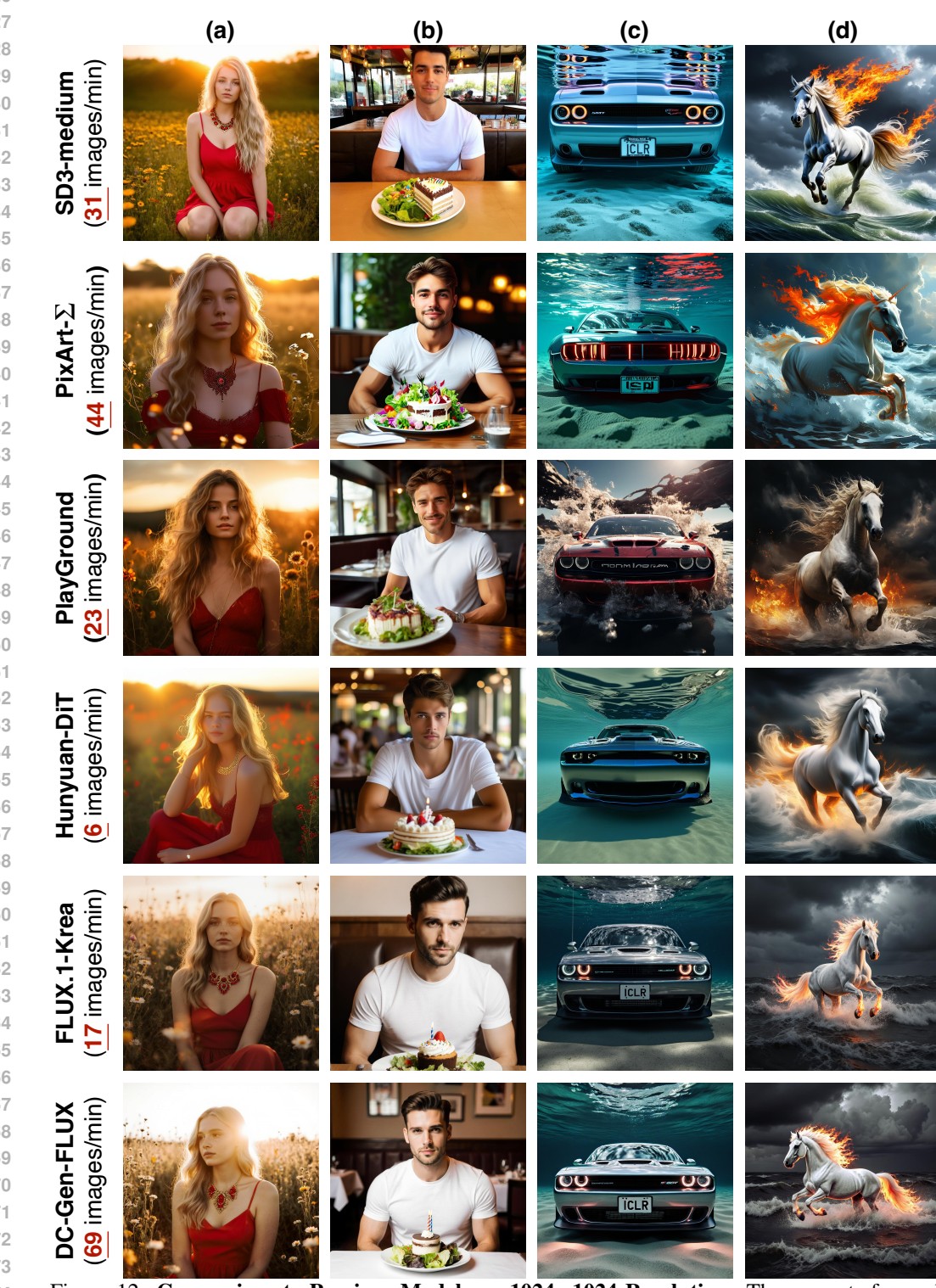

Figure 12: **Comparison to Previous Models on 1024×1024 Resolution.** The prompts for each column are: **(a)** A young woman in a red dress sits peacefully in a field of wildflowers, her long blonde hair glowing in the golden sunlight. **(b)** A handsome man in a white T-shirt sits at a restaurant table with a salad topped with birthday cake. **(c)** An 3D render of a Dodge Challenger SRT Hellcat with 'ICLR' license plate, submerged on a sandy ocean floor. **(d)** A powerful white horse with a fiery mane gallops through dark, churning waves, emanating light against a dramatic sky.

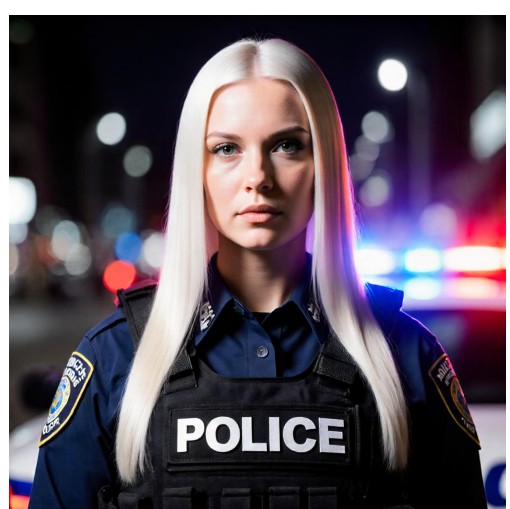

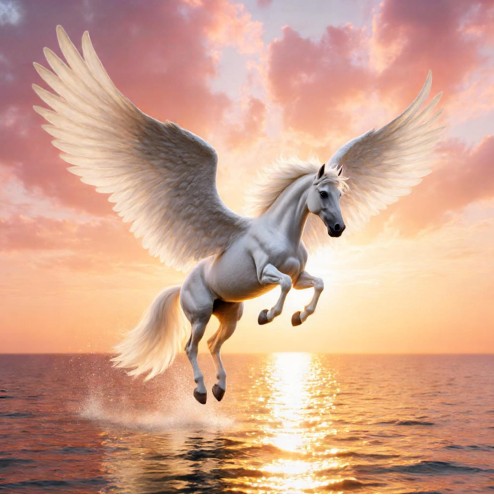

**Prompt:** Long white-haired female officer in a dark blue uniform and tactical vest with "POLICE" lettering. The background is a blurry urban backdrop with flashing police car lights.

**Prompt:** A majestic white Pegasus soars powerfully mid-flight, wings spread wide against a breathtaking sunset. The sky's vibrant orange and pink hues reflect off the shimmering ocean below.

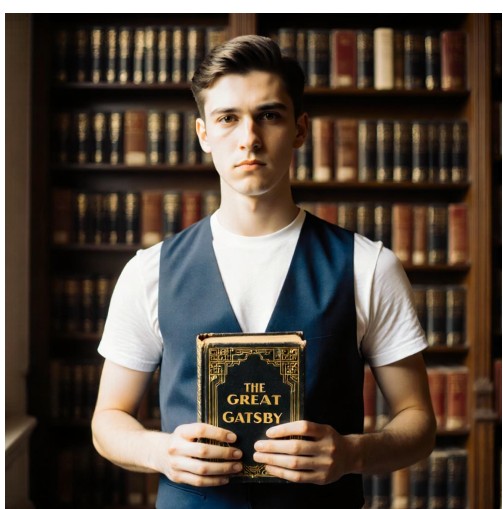

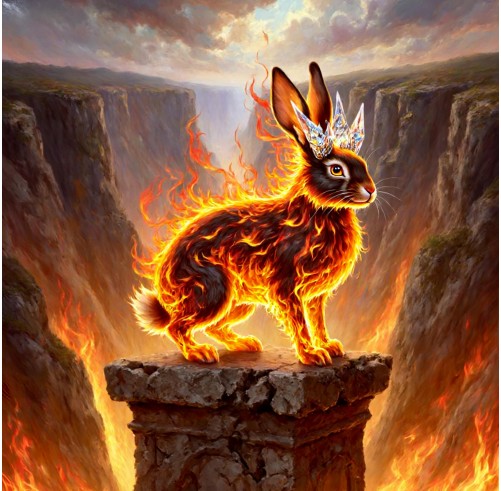

**Prompt:** A young man with serious expression, stands in a library, holding "The Great Gatsby." He wears simple white tee, and blue vest.

**Prompt:** A majestic fire rabbit with crystalline horns stands on a pedestal, overlooking a fiery chasm. Its body is wreathed in magical, shifting flames.

Figure 13: **Samples of DC-Gen-FLUX on 4096×4096 Resolution.** The base model FLUX.1-Krea does not support native 4K image generation as shown Fig. 7. While DC-Gen-FLUX benefits from the deeply compressed latent space to enable efficient training on high-resolution images, thereby unlocking native 4K generation capability.

## A.5 Additional Human Evaluation Details

**Instructions**

Review the two image options and the text description, then answer the following questions:

- **Quality Comparison:** Select the image option that shows better visual quality. (You may choose "similar visual quality" if you think their visual qualities are similar.)
- **Text Alignment:** Select the image option that better aligns the given text description. (You may choose "similar text alignment" if you think their text alignments are similar.)

**Text Description:** "A well-groomed man with short brown hair and glasses smiles warmly in a professional portrait. He wears a dark navy suit, white shirt, and red tie against a blurred blue-gray background, conveying confidence, approachability, and political campaign readiness."

**Option 1**

**Option 2**

**Question**

**1. Select the image option that shows better visual quality.**

○ Option 1    ○ Option 2

○ Similar Visual Quality

**2. Select the image option that better aligns the given text description.**

○ Option 1    ○ Option 2

○ Similar Text Alignment

Submit

Figure 14: **Illustration of the Human Evaluation Interface.**

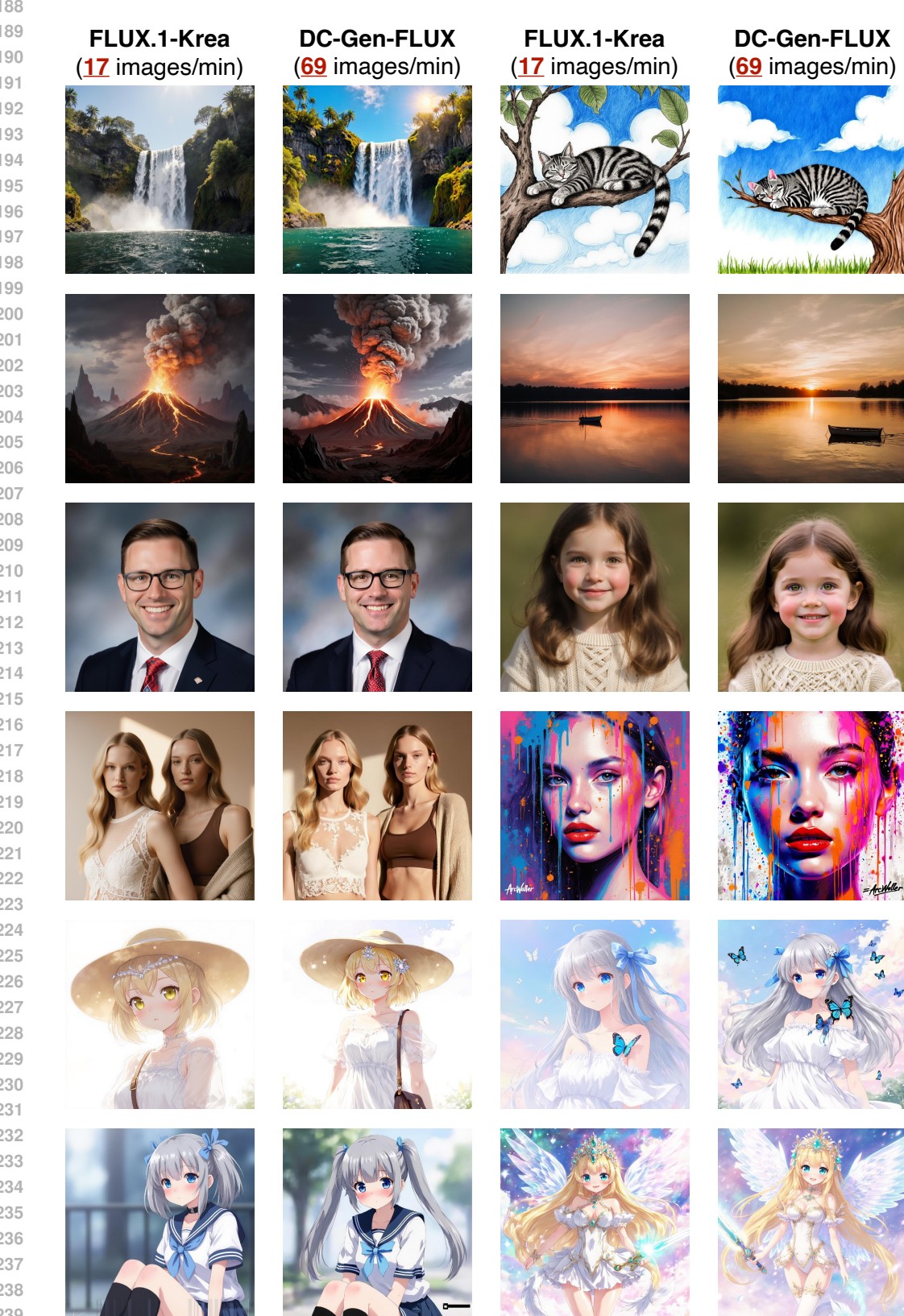

Figure 15: **Human Evaluation Sample Image Pairs.**

## A.6 ADDITIONAL 4K-RESOLUTION IMAGE GENERATION RESULTS

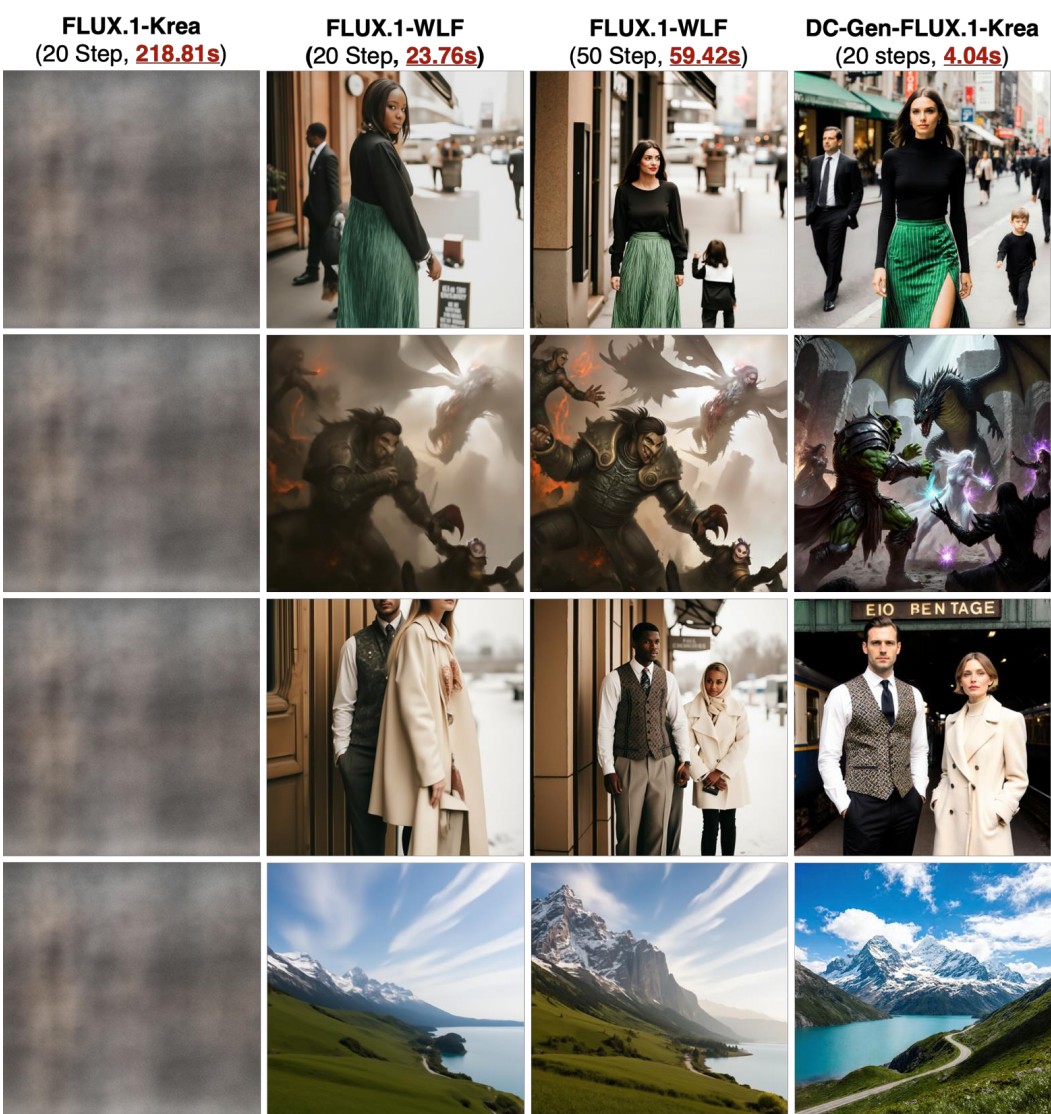

Figure 16: **Visual Results on the Diffusion-4K Benchmark.**

## A.7 AUTOENCODER RECONSTRUCTION RESULTS ON HIGH-RESOLUTION IMAGES

DC-Gen focuses on high-resolution image generation, which naturally contains substantial spatial redundancy. Deeply compressed autoencoders such as DC-AE (Chen et al., 2024c) leverage this redundancy to reduce token counts without sacrificing image quality. Fig. 17 illustrates reconstruction results from DC-AE-f32c32 on high-resolution input images. The model faithfully preserves fine-grained details, such as small human faces, text, and subtle shadows.

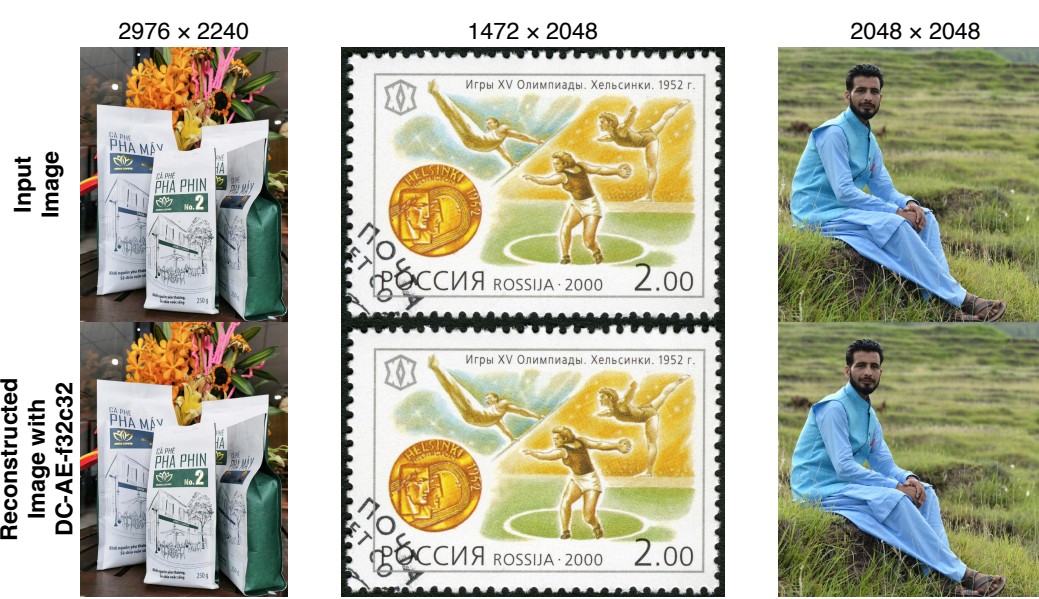

Figure 17: **Illustration of Autoencoder Reconstruction Results on High-Resolution Images.**

## A.8 COMPARISON WITH ADDITIONAL ADAPTATION STRATEGIES

In addition to the extensive comparisons with direct end-to-end fine-tuning reported in Table. 8, we further compare DC-Gen with three additional adaptation strategies:

| Text-to-Image Generation Results on MJHQ-30K 512×512 | | | |
|---|---|---|---|
| **Model and Autoencoder** | **Method** | **FID ↓** | **CLIP-Score ↑** |
| SANA-1.6B
(DC-AE-f32c32 → DC-AE-f64c128) | Freeze-Backbone Then End-to-End | 630.69 | 14.90 |
| | Layer-Wise LR Decay | 407.26 | 14.06 |
| | Gradual LoRA | 11.61 | 25.07 |
| | DC-Gen | **5.10** | **28.04** |
| FLUX.1-Krea-12B
(FLUX-VAE-f8c16 → DC-AE-f32c32) | Freeze-Backbone Then End-to-End | 16.02 | 26.58 |
| | Layer-Wise LR Decay | 17.83 | 25.77 |
| | Gradual LoRA | 15.90 | 26.52 |
| | DC-Gen | **13.30** | **27.18** |

Table 9: **Quantitative Comparison with Additional Adaptation Strategies.**

- **Freeze-Backbone Then End-to-End.** In this baseline, we first freeze the DiT backbone and train only the patch embedding layer and output head, followed by full end-to-end training with LoRA.
- **Layer-Wise Learning Rate Decay.** Instead of using a uniform learning rate, we apply a linear layer-wise learning rate decay, where earlier layers receive smaller learning rates and later layers

receive larger ones, while keeping the average learning rate unchanged. The ratio between the smallest and largest learning rates is 1:3.

- **Gradually Introducing LoRA Modules.** In this baseline, we use a linear warmup schedule for the LoRA alpha value, starting at 0 and increasing linearly to the target value.

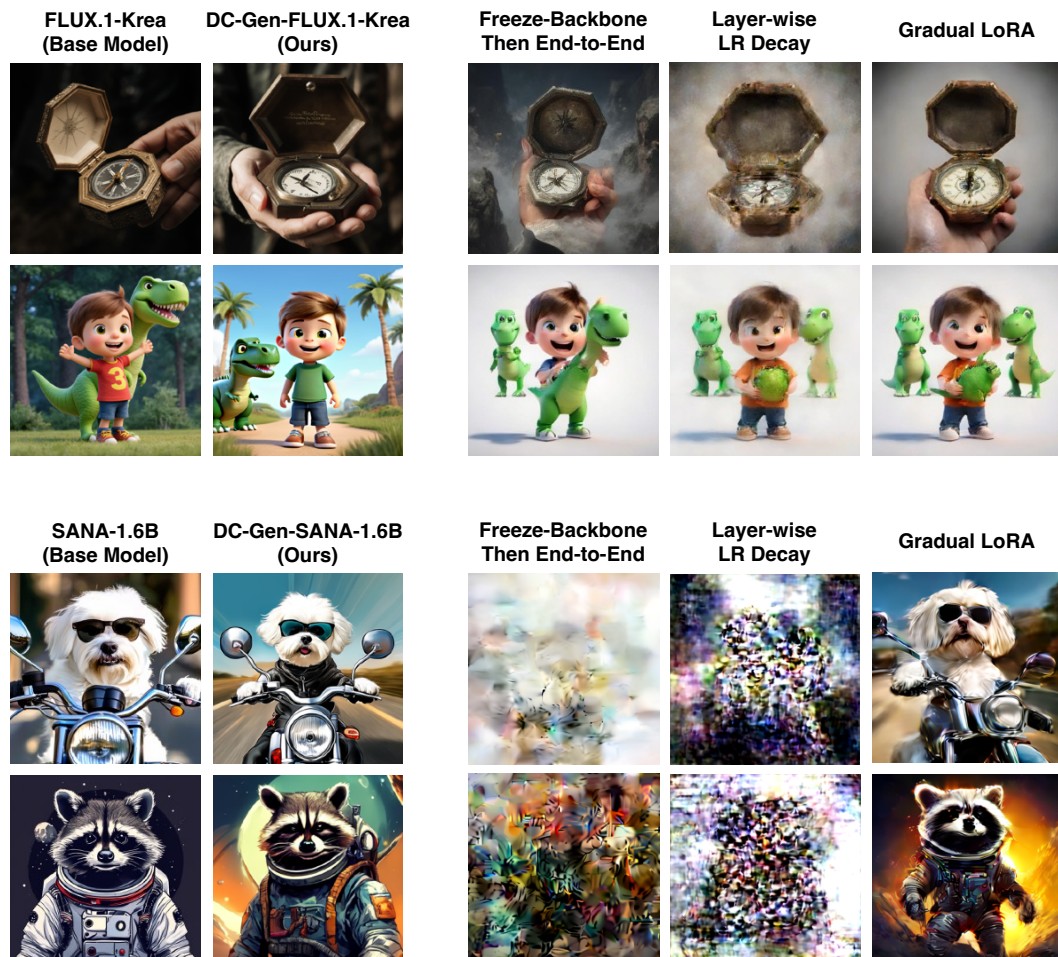

Figure 18: **Visual Comparison with Additional Adaptation Strategies.**

Table. 9 and Fig. 18 summarize the results. DC-Gen outperforms all adaptation strategies by a significant margin, achieving higher CLIP scores, better FIDs, and noticeably better visual quality.

## A.9 COMPARISON WITH AUTOENCODER LATENT SPACE ALIGNMENT

| Text-to-Image Generation Results on MJHQ-30K 512×512 | | | |
|---|---|---|---|
| **Method** | **AE Alignment Ratio** | **gFID ↓** | **CLIP-Score ↑** |
| Autoencoder Latent Space Alignment | 0.1 | 15.39 | 26.53 |
| | 1.0 | 15.27 | 26.36 |
| | 10.0 | 21.76 | 23.20 |
| DC-Gen | - | **13.30** | **27.18** |

Table 10: **Comparison with Autoencoder Latent Space Alignment.** The base model is FLUX.1-Krea-12B.

We compare DC-Gen with Autoencoder Latent Space Alignment. Specifically, we fine-tune the DC-AE model (Chen et al., 2024c) using an additional alignment loss that encourages its latent space to match that of FLUX-VAE-f8c16. We experiment with three alignment loss ratios—0.1, 1.0, and 10.0. As shown in Table. 10, DC-Gen achieves substantially better results compared to Autoencoder Latent Space Alignment.

## A.10 GRADIENT NORM CURVE

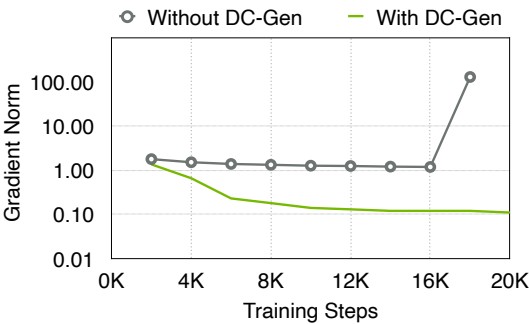

Figure 19: **Gradient Norm Curve.** The base model is SANA-1.6B.

Fig. 19 presents the gradient norm curve on SANA-1.6B. The gradient norm of the vanilla adaptation is substantially higher than that of DC-Gen, and it further exhibits clear gradient explosion around step 16K.

## A.11 ADDITIONAL VISUAL RESULTS OF DC-GEN

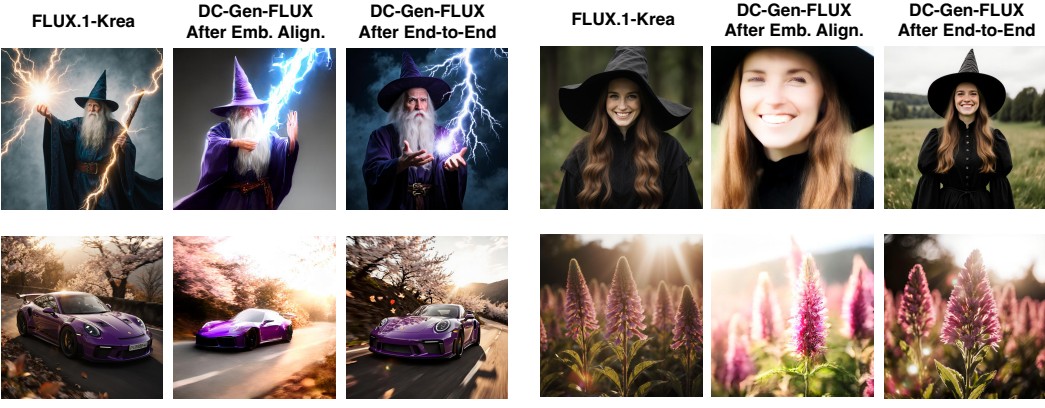

Figure 20: **Additional Visual Results of DC-Gen.**

