# OpenReview forum: "DC-Gen: Post-Training Diffusion Acceleration with Deeply Compressed Latent Space"
_ICLR.cc/2026/Conference — Submitted to ICLR 2026_

### Official Review · Reviewer_UXr9 · 2025-10-25

**Soundness:** 2
**Presentation:** 3
**Contribution:** 2
**Rating:** 4
**Confidence:** 5

**Summary:**

This paper proposes DC-GEN, a framework for accelerating large diffusion models by using a deeply compressed latent space. The authors introduce two stages: (1) embedding alignment, where the patch embedder of the diffusion model is fine-tuned to match the new latent representation; and (2) LoRA fine-tuning, used to adapt the diffusion backbone in the new latent space. They report large acceleration ratios (up to 50×) with minimal visual degradation in image generation quality.

**Strengths:**

1. Clear and practical motivation: The goal—reducing diffusion inference cost for large-scale image synthesis—is timely and relevant. The focus on post-training acceleration (rather than retraining) is appealing for efficiency-oriented applications.

2. Method is clearly explained: The embedding alignment and LoRA adaptation procedures are easy to follow, and the pipeline is well illustrated.

3. Empirical evaluation is adequate: The paper shows decent acceleration on high-resolution image generation and uses well-known benchmarks for comparison.

**Weaknesses:**

1. Limited novelty. The proposed framework largely combines known components: A standard embedding alignment step (MSE between pretrained and downsampled embeddings); LoRA fine-tuning on synthetic data. These are conventional adaptation techniques and do not introduce a fundamentally new concept or architecture.

2. “Post-training” claim is misleading. The process still requires fine-tuning a model on paired data, sometimes with significant compute and dataset size. This is not truly “post-training” in the sense of zero-cost adaptation.

3. Limited practicality for large-scale models. Although LoRA reduces the fine-tuning cost, successfully matching the teacher model’s performance still requires access to the original training recipe—including comparable data quality, preprocessing, and post-training optimization settings. Without these, the adapted model inevitably falls short. Using synthetic data from the teacher can lower the data requirement but introduces another issue: the model may inherit and amplify the teacher’s artifacts and stylistic biases within the compressed latent space.

**Questions:**

Why did the authors choose to retrain the patch embedder rather than adapt the DC-VAE latent space directly to the pretrained VAE? Would aligning the autoencoder latent representation yield more consistent features with less fine-tuning?

Does the high-compression autoencoder introduce any artifacts or perceptual degradation in the generated images after adaptation? If so, how are these quantified or mitigated?

Why does the model appear to achieve better quantitative scores after switching to a high-compression autoencoder?

---

> ### Author Response · Authors · 2025-11-24
>
> ### Q1: Novelty and Technical Contributions
>
> We respectfully disagree. DC-Gen’s key contribution lies in identifying and addressing a fundamentally new challenge in adapting pre-trained diffusion models to deeply compressed latent spaces. This work opens a new direction for accelerating pre-trained diffusion models through autoencoder replacement, while prior works mainly relied on sparsity, quantization, caching methods, etc.
>
> Prior work has not recognized that the core issue is the severe mismatch in the embedding space when changing autoencoders. DC-Gen reveals this new insight and provides a principled, domain-specific adaptation strategy that effectively resolves the issue. The table below summarizes the comparison with other adaptation strategies (more details are provided in Appendix A.8 and Appendix A.3.1).
>
> |Base Model|Method|FID|CLIP-Score|
> |-|-|-|-|
> |SANA-1.6B (DC-AE-f32c32 -> DC-AE-f64c128) | Freeze-Backbone Then End-to-End | 630.69  | 14.90 |
> | | Direct End-to-End | 258.50 | 13.72 |
> | | Layer-Wise LR Decay  | 407.26 | 14.06 |
> | | Gradual LoRA | 11.61 | 25.07 |
> | | **DC-Gen** | **5.10** | **28.04** |
> | FLUX.1-Krea-12B (FLUX-VAE-f8c16 -> DC-AE-f32c32) | Freeze-Backbone Then End-to-End | 16.02 | 26.58 |
> | | Direct End-to-End | 15.78 | 26.50 |
> | | Layer-Wise LR Decay  | 17.83 | 25.77 |
> | | Gradual LoRA | 15.90 | 26.52 |
> | | **DC-Gen** | **13.30** | **27.18** |
>
> DC-Gen outperforms all other adaptation strategies by a substantial margin and delivers noticeably better visual results (Appendix Figure 18 and Figure 9).
>
> ### Q2: Limited Practicality for Large-Scale Models
>
> The focus of DC-Gen is accelerating pre-trained diffusion models while preserving high-resolution image generation quality. We provide strong empirical evidence that DC-Gen can match the quality of the large-scale diffusion model without access to its original training recipe (e.g., FLUX.1-Krea only released its inference code and model weights), as demonstrated by extensive quantitative (Table 2), qualitative (Figure 11, Figure 15), and human evaluation (Table 4) comparisons with FLUX.1-Krea-12B.
>
> Avoiding the inheritance of potential artifacts or stylistic biases from the base model is beyond the scope of this work, as our objective is to closely mimic the base model’s behavior.
>
> ### Q3: "Post-training claim is misleading"
>
> Post-training is a general term for processes applied after a model’s main training has been completed. It does not necessarily imply zero cost; for example, in large language models, instruction tuning and reinforcement learning training are also considered part of post-training.
>
> In our case, since we start from a pre-trained diffusion model rather than training from scratch, post-training is fully consistent with our setting.
>
> ### Q4: Comparison with Autoencoder Latent Space Alignment
>
> We compare DC-Gen with Autoencoder Latent Space Alignment in Section A.9. Specifically, we fine-tune the DC-AE model using an additional alignment loss that encourages its latent space to match that of FLUX-VAE-f8c16. We experiment with three alignment loss ratios—0.1, 1.0, and 10.0. As shown in table below, DC-Gen achieves substantially better results compared to Autoencoder Latent Space Alignment.
>
> | Method | AE Alignment Ratio  | FID | CLIP-Score |
> |-|-|-|-|
> |Autoencoder Latent Space Alignment| 0.1 | 15.39 | 26.53 |
> | | 1.0 | 15.27 | 26.36 |
> | | 10.0 | 21.76 | 23.20 |
> | **DC-Gen** | - | **13.30** | **27.18** |
>
> ### Q5: Does the high-compression autoencoder introduce any artifacts or perceptual degradation?
>
> DC-Gen focuses on high-resolution image generation, which naturally contains substantial spatial redundancy. Deeply compressed autoencoders such as DC-AE leverage this redundancy to reduce token counts without sacrificing image quality. Appendix Figure 17 illustrates reconstruction results from DC-AE-f32c32 on high-resolution input images. The model faithfully preserves fine-grained details, such as small human faces, text, and subtle shadows.
>
> Appendix Figure 15 and Table 4 summarize our human evaluation results. We do not observe any perceptual degradation in the generated images after adaptation.
>
>
> ### Q6: Why does the model appear to achieve better quantitative scores?
>
> A high-compression autoencoder can exploit spatial redundancy to reduce token counts without compromising image quality for high-resolution images. Using such an autoencoder does not necessarily degrade performance; for example, [1] reports that using a high-compression autoencoder achieves better ImageNet FID than using an 8× spatial compression autoencoder.
>
> [1] Chen, Junyu, et al. "Deep compression autoencoder for efficient high-resolution diffusion models." ICLR (2025).

---

### Official Review · Reviewer_Zszk · 2025-11-01

**Soundness:** 2
**Presentation:** 4
**Contribution:** 2
**Rating:** 4
**Confidence:** 5

**Summary:**

This paper proposes DC-Gen, a post-training framework to accelerate pretrained text-to-image diffusion models by adapting them to a deeply compressed latent space (e.g., moving from 8× to 32×/64× autoencoder compression).
The approach introduces two stages: Embedding Alignment, which aligns the pretrained model’s latent embeddings with those of a new DC-AE autoencoder to stabilize adaptation, and LoRA Fine-Tuning which lightly fine-tunes the model to the new latent space, preserving semantics while being much cheaper.
The method is validated on SANA and FLUX.1-Krea, showing comparable visual quality and FID/CLIP scores to base models, while achieving 53× latency reduction for 4K generation on H100 GPUs.
DC-Gen shrinks the adaptation time to only 40 H100 GPU-days.

**Strengths:**

1. The paper is clearly written and easy to follow.

2. The demonstrated results are promising. The authors provide a comprehensive study on visual results, quantitative metrics, and speed benchmarks. The speed gain and 4K generation quality are impressive.

3. This work addresses an important research topic in a timely manner, i.e., accelerating diffusion generation through adapting the denoiser to a highly compressed latent space.

**Weaknesses:**

1. Novelty/technical contribution is my biggest concern. Freezing the main body (DiT blocks) of the denoiser and finetuning the projection layers to adapt to a new VAE is a reasonable but very trivial and common technique. I am not surprised that this will show faster convergence compared to simple full finetuning.

2. I have some concerns about the baseline and comparison. The scope of DC-Gen is an efficient adaptation method for a high compression autoencoder. But most of the results are comparing baseline models. From my point of view, the most important comparison is against full finetuning/adaptation, but the authors claim this leads to model collapse. Please refer to my Q1. In the main comparisons, DC-Gen achieves a comparable quality compared to FLUX or SANA, while being much faster. But I would attribute this gain more to the high-compression VAE instead of the adaptation method.

**Questions:**

1. Why does vanilla adaptation fail as in Fig.3? Can you provide a more detailed analysis? Normally, diffusion training is more stable compared to previous methods, such as GAN. It is a bit surprising to see that the model is already starting to adapt to the new latent space, but just suddenly collapses. Is it because of gradient explosion?

2. Can authors provide a latency breakdown for 4K generation? Because of the highly compressed latent space, the denoiser speed is successfully managed in this work, but I am curious about the VAE decoding cost.

---

> ### Author Response · Authors · 2025-11-24
>
> ### Q1: Novelty/Technical Contributions and Additional Baselines
>
> There appears to be a misunderstanding. DC-Gen is not equivalent to simply freezing the DiT blocks and fine-tuning the projection layers. Instead, we identify a fundamental challenge: a severe mismatch in the embedding space after changing the autoencoder. Building on this novel insight, we introduce a principled, domain-specific adaptation strategy that addresses this issue through a dedicated embedding-alignment stage, which trains the patch embedder to mimic the base model’s embedding space. Without our novel design, all alternative adaptation strategies exhibit substantial quality degradation, including full fine-tuning (denoted as Direct End-to-End) and the approach of fine-tuning the projection layers with a frozen DiT backbone followed by end-to-end training (denoted as Freeze-Backbone Then End-to-End).
>
> |Base Model|Method|FID|CLIP-Score|
> |-|-|-|-|
> |SANA-1.6B (DC-AE-f32c32 -> DC-AE-f64c128) | Freeze-Backbone Then End-to-End | 630.69  | 14.90 |
> | | Direct End-to-End | 258.50 | 13.72 |
> | | Layer-Wise LR Decay  | 407.26 | 14.06 |
> | | Gradual LoRA | 11.61 | 25.07 |
> | | **DC-Gen** | **5.10** | **28.04** |
> | FLUX.1-Krea-12B (FLUX-VAE-f8c16 -> DC-AE-f32c32) | Freeze-Backbone Then End-to-End | 16.02 | 26.58 |
> | | Direct End-to-End | 15.78 | 26.50 |
> | | Layer-Wise LR Decay  | 17.83 | 25.77 |
> | | Gradual LoRA | 15.90 | 26.52 |
> | | **DC-Gen** | **13.30** | **27.18** |
>
> Further details are provided in Appendix A.8 and Appendix A.3.1, with visual results shown in Appendix Figures 18 and 9. To the best of our knowledge, this is the first method that enables adapting pretrained diffusion models to a deeply compressed latent space without quality degradation. This capability opens a new direction for accelerating pretrained diffusion models through autoencoder replacement.
> ### Q2: Why does vanilla adaptation fail?
>
> As discussed in Section 3.2, we hypothesize that the root cause is the embedding-space mismatch. Beginning with a poor initialization—namely, a randomly initialized patch-embedding layer combined with a pretrained DiT backbone—makes training notably less stable. Appendix Figure 19 shows the gradient-norm curve on SANA-1.6B: the gradient norm under the vanilla adaptation strategy is substantially higher than that of DC-Gen and exhibits a clear gradient explosion around step 16K.
>
> Even when gradient explosion does not occur, the method still experiences a significant quality drop, as demonstrated in our experiments on FLUX.1-Krea-12B.
>
> ### Q3: Latency Breakdown for 4K Generation
>
> The latency breakdown for 4K generation (measured on the NVIDIA H100 GPU) is summarized below:
>
> |Model | Diffusion Model Latency (s) | AE Decoding Latency (s) | Total (s) |
> |-|-|-|-|
> | Flux.1-Krea-12B | 213.81 | 5.58 | 219.39 |
> | DC-Gen-Flux.1-Krea-12B | 4.04 | 0.95 | 4.99 |

---

### Official Review · Reviewer_mkD3 · 2025-11-01

**Soundness:** 3
**Presentation:** 3
**Contribution:** 3
**Rating:** 6
**Confidence:** 3

**Summary:**

The paper designs a post training method to accelerate existing text-to-image diffusion models. The argues that that image generation at 4K resolution for example becomes computationally expensive due to inefficient encoding in the text-to-image diffusion models. While training new models from scratch on compressed latent spaces is expensive, the paper employes Embedding Alignment by training only the patch embedder of the model and output head, using a loss that on downsampled features from the base embedder. Stage 2 employs a lightweight LoRA training for the final alignment. This method is applied to SANA and FLUX.1-Krea models. The resulting DC-Gen-FLUX achieves a 53x latency reduction for 4K image generation on an H100 GPU. The authors claim that the reduction in training cost with respect to training the models from scratch is huge.

**Strengths:**

1) The paper provides a way to post train diffusion models to support higher resolution while maintaining the prior. The model is able to generate 4K images on lower budget.

2) The paper claims  a 53x speedup in latency for 4K generation.

3) The paper employs a two stage approach where first the model representations are aligned and then adapted. This is intuitive

4) The results show that the model is able to preserve the prior of the model from which the model is distilled from.

**Weaknesses:**

1) While the method aligns with a particular encoder, the inherent low quality reconstruction of highly compressed models still remains a problem.

2) The paper simplifies downsampling to a simple, non-parametric operation. This method may not adequately handle high-frequency details from the original latent space. This might blur our some details.

3) Human eval is needed as FID and CLIP scores are too coarse. It is unclear if the blurriness is solved.

4) While the paper claims to operate at 4K resolution. There is no evaluation for this quality. Training on small set of 4K images could be evaluated to support the claim.

**Questions:**

I have questions mainly focussed on weaknesses:

1) How do the authors defend the inherent low quality of highly compressed latent spaces?

2) Since the downsampling is not optimal, what alternative methods can preserve the frequencies ? or given a downsampling budget what is the best way to do it , this is not clear.

3) How reliable are FID and CLIP , do they detect the blurriness in this case? how sensitive are these metrics to this artifact?

4) What about the evaluation at 4K resolution? The model can generate this resolution but how to evaluate the quality? This part is missing.

---

> ### Author Response · Authors · 2025-11-24
>
> ### Q1: The inherently low reconstruction quality of highly compressed autoencoder models.
>
> DC-Gen focuses on high-resolution image generation, which naturally contains substantial spatial redundancy. Deeply compressed autoencoders such as DC-AE leverage this redundancy to reduce token counts without sacrificing image quality. Appendix Figure 17 illustrates reconstruction results from DC-AE-f32c32 on high-resolution input images. The model faithfully preserves fine-grained details, such as small human faces, text, and subtle shadows.
>
> ### Q2: Why use a simple, non-parametric downsampling operation for Embedding Alignment?
>
> In our experiments, this simple non-parametric downsampling operation for embedding alignment is sufficient to achieve strong results. We believe this is because the alignment stage serves only as the first step of our pipeline that ensures training stability in the following processes, and the subsequent end-to-end fine-tuning stage enables the model to recover fine details, as demonstrated in Appendix A.11.
>
> Since this minimalist design already performs well, we opt to retain it to keep the overall framework as simple as possible.
> ### Q3: Human Evaluation
>
> We conducted a human evaluation comparing DC-Gen-FLUX.1-Krea and FLUX.1-Krea on 1K-resolution image generation. To ensure an unbiased assessment, we used Amazon Mechanical Turk, a crowdsourcing platform. Full details are provided in Section 4.3. DC-Gen-FLUX.1-Krea achieves visual quality and text alignment comparable to FLUX.1-Krea, with a slightly higher win rate across all domains.
>
> ### Q4: 4K-Resolution Evaluation
>
> Following your suggestion, we added a 4K-Resolution Image Generation Evaluation in Section 4.2.2. FLUX.1-Krea does not support native 4K image generation. In addition to our approach, [1] provides a state-of-the-art framework for enabling 4K generation. DC-Gen offers substantially better efficiency while also achieving higher CLIP scores, higher Aesthetic scores, and noticeably better visual quality.
>
> [1] Zhang, Jinjin, et al. "Diffusion-4k: Ultra-high-resolution image synthesis with latent diffusion models." Proceedings of the Computer Vision and Pattern Recognition Conference. 2025.

---

### Official Review · Reviewer_UuTi · 2025-11-02

**Soundness:** 4
**Presentation:** 4
**Contribution:** 2
**Rating:** 6
**Confidence:** 4

**Summary:**

The paper proposes DC-Gen, which is a 3-stage pipeline that adapts a pre-trained latent diffusion model to a different VAE. In the first embedding alignment stage, a new patch embedder is randomly initialized and then aligned with the original embedder (w/ the original VAE encoder) by minimizing an MSE loss. In the second output head alignment stage, the randomly-initialized output head are jointly trained with the embedder from stage 1. In the third stage, LoRA modules are attached to backbone transformer blocks and jointly trained using the flow matching loss. The progressive adaptation ensures that the model converges stably while maintaining a relatively low training budget. The pipeline is evaluated on ImageNet DiT, SANA and FLUX, and demonstrates convincing results under all settings.

**Strengths:**

* The paper is well written and easy to follow, with a clear structure and most of the important details adequately covered.

* The empirical results are solid, with both generation quality and speedup demonstrated under a broad range of settings (different models, class-conditioned and text-conditioned image generation, and multiple datasets).

* It is somewhat interesting to observe that LDM backbones appear to retain certain latent-space-agnostic knowledge, which can be efficiently transferred across vastly different VAEs.

**Weaknesses:**

* The novelty appears to be somewhat limited. The general idea of training newly added layers first as an alignment has been a common practice for a long time and is ubiquitous in many similar use cases, e.g., appending visual encoders to the input of LLMs [1, 2]. Therefore, although this work may be among the first to highlight the significance of alignment when changing VAE in LDMs, the overall training pipeline itself appears relatively straightforward to derive.

* Following the previous point, several well-established or straightforward alternatives, such as layer-wise learning rate decay or gradually introducing LoRA modules through gating mechanism (zero-initialized and learned / following a predefined schedule), also seem to be possible ways to mitigate the interference of random weights in the early stage of the training, as is observed in the paper. It might make the advantage of the proposed method a stronger argument if more baselines could be considered (via experiments or analysis) beyond direct finetuning or training from scratch.

[1] Liu, Haotian, et al. "Visual instruction tuning." Advances in neural information processing systems 36 (2023): 34892-34916.

[2] Bai, Jinze, et al. "Qwen-vl: A frontier large vision-language model with versatile abilities." arXiv preprint arXiv:2308.12966 1.2 (2023): 3.

**Questions:**

* Ln 291-294 and Appendix Table 5 mentioned a separate output head alignment stage. What is the loss function used in this stage? It might be helpful to include the output head alignment stage as part of Fig. 2(b).

* At around Ln 414, how exactly is the synthetic dataset generated (e.g., from a fixed set of prompts, or generated unconditionally)?

* In Table 5, could the authors provide the wall-clock GPU hours for each training stage? Since the training time may not be proportional with the number of training steps due to the differences in training settings, this information would help identify potential bottlenecks in the proposed pipeline and in turn guide optimizations in future works.

* How are the position embeddings handled after the input shape changes?

---

> ### Author Response · Authors · 2025-11-24
>
> ### Q1: Novelty/Technical Contributions and Additional Baselines
>
> DC-Gen’s key contribution lies in identifying and addressing a fundamentally new challenge in adapting pre-trained diffusion models to deeply compressed latent spaces. This work opens a new direction for accelerating pre-trained diffusion models through autoencoder replacement, while prior works mainly relied on sparsity, quantization, caching methods, etc.
>
> Prior work has not recognized that the core issue is the severe mismatch in the embedding space when changing autoencoders. DC-Gen reveals this new insight and provides a principled, domain-specific adaptation strategy that effectively resolves the issue. The table below summarizes the comparison with other adaptation strategies (more details are provided in Appendix A.8 and Appendix A.3.1).
>
> |Base Model|Method|FID|CLIP-Score|
> |-|-|-|-|
> |SANA-1.6B (DC-AE-f32c32 -> DC-AE-f64c128) | Freeze-Backbone Then End-to-End | 630.69  | 14.90 |
> | | Direct End-to-End | 258.50 | 13.72 |
> | | Layer-Wise LR Decay  | 407.26 | 14.06 |
> | | Gradual LoRA | 11.61 | 25.07 |
> | | **DC-Gen** | **5.10** | **28.04** |
> | FLUX.1-Krea-12B (FLUX-VAE-f8c16 -> DC-AE-f32c32) | Freeze-Backbone Then End-to-End | 16.02 | 26.58 |
> | | Direct End-to-End | 15.78 | 26.50 |
> | | Layer-Wise LR Decay  | 17.83 | 25.77 |
> | | Gradual LoRA | 15.90 | 26.52 |
> | | **DC-Gen** | **13.30** | **27.18** |
>
> DC-Gen outperforms all other adaptation strategies by a substantial margin and delivers noticeably better visual results (Appendix Figure 18 and Figure 9).
> ### Q2: How was the synthetic dataset generated?
>
> We generated the synthetic dataset by providing a fixed set of text prompts to the base model, employing classifier-free guidance during inference.
> ### Q3: GPU Wall-Clock Time for Each Training Stage
>
> We summarize the GPU wall-clock time for each training stage below:
>
>
> |Base Model|Patch Embedder Alignment (H100 GPU Days) | Output Head Alignment (H100 GPU Days)|End-to-End Fine-Tuning (H100 GPU Days)|
> |-|-|-|-|
> |DiT-XL|0.023|0.060|0.448|
> |SANA-1.6B|0.037|0.593|26.637|
> |SANA-4.8B|0.037|1.305|19.583|
> |FLUX.1-Krea-12B|0.084|9.708|28.417|
>
> ### Q4: Position Embeddings After Input Shape Changes
>
> After the input shape changed, we discarded the original positional embedding and rebuilt a new one for the updated shape, following the identical way as the base model.
>
> ### Q5: Loss Function for Output Head Alignment
>
> The loss function is identical to that used in the end-to-end fine-tuning stage. We have revised the caption of Figure 2 and Table 7 to make it more clear.

---

### Author Response · Authors · 2025-11-24

We sincerely thank all reviewers for their constructive comments. Our manuscript has been updated accordingly, with changes highlighted in gold. Please see our point-by-point responses below each review. We greatly appreciate your feedback and welcome any further questions!

---

### Meta-Review · Area_Chair_mnz4 · 2026-01-09

**Summary:**

This paper accelerates pre-trained diffusion models mainly by replacing the autoencoder, addressing an important challenge for diffusion transformer models. Despite the relevance of the problem, two reviewers gave positive scores (6 from UuTi and mkD3), while the other two provided negative scores (4 from Zszk and UXr9).

**Reviewer Concerns:**

Most reviewers raised concerns regarding the novelty (alignment has been widely used, and the conventional adaptation technique LoRA), the methodology (the use of highly compressed latent spaces leading to low-quality reconstructions), as well as the evaluations (e.g., FID and CLIP). During the rebuttal period, the authors attempted to address these issues, but their responses did not appear to satisfy the reviewers.

The AC also agrees with the reviewers’ concerns, particularly those regarding the novelty of the method. Therefore, I recommend rejecting this paper.

**Reviewer Scores:**

The reviewers are likely to maintain their scores.

---

### Decision · Program_Chairs · 2026-01-26

Reject